# LEARNING DEEP OPERATOR NETWORKS: THE BENEFITS OF OVER-PARAMETERIZATION

## ABSTRACT

Neural Operators that directly learn mappings between function spaces have received considerable recent attention. Deep Operator Networks (DeepONets) (Lu et al., 2021), a popular recent class of operator networks have shown promising preliminary results in approximating solution operators of parametric partial differential equations. Despite the universal approximation guarantees (Lu et al., 2021; Chen & Chen, 1995) there is yet no optimization convergence guarantee for DeepONets based on gradient descent (GD). In this paper, we establish such guarantees and show that over-parameterization based on wide layers provably helps. In particular, we present two types of optimization convergence analysis: first, for smooth activations, we bound the spectral norm of the Hessian of DeepONets and use the bound to show geometric convergence of GD based on restricted strong convexity (RSC); and second, for ReLU activations, we show the neural tangent kernel (NTK) of DeepONets at initialization is positive definite, which can be used with the standard NTK analysis to imply geometric convergence. Further, we present empirical results on three canonical operator learning problems: Antiderivative, Diffusion-Reaction equation, and Burger's equation, and show that wider DeepONets lead to lower training loss on all the problems, thereby supporting the theoretical results.

## 1 INTRODUCTION

Replicating the success of Deep Learning in scientific computing such as developing Neural PDE solvers, constructing surrogate models and developing hybrid numerical solvers has recently captured interest of the broader scientific community. Neural Operators (Li et al., 2021a;b) and Deep Operator Networks (DeepONets) (Lu et al., 2021; Wang et al., 2021) encompass two recent approaches aimed at learning mappings between function spaces. Contrary to a classical supervised learning setup which aims at learning mappings between two finite-dimensional vector spaces, these neural operators/operator networks aim to learn mappings between *infinite-dimensional* function spaces. The key underlying idea in both the approaches is to parameterize the solution operator as a deep neural network and proceed with learning as in a standard supervised learning setup. Since a neural operator directly learns the mapping between the input and output function spaces, it is a natural choice for learning solution operators of parametric PDE's where the PDE solution needs to be inferred for multiple instances of these "input parameters" or in the case of inverse problems when the forward problem needs to be solved multiple times to optimize a given functional. While there exist results on the approximation properties and convergence of DeepONets; see, e.g., (Deng et al., 2021) for a convergence analysis of DeepONets – vis-a-vis their approximation guarantees– for the advection-diffusion equation, there do not exist any optimization results on when and why GD converges during the optimization of the DeepONet loss.

In this work we put forth theoretical convergence guarantees for DeepONets centered around over-parameterization and show that over-parameterization based on wider layers (for both branch and trunk net) provably helps in DeepONet convergence. This is reflected in Figure 1 which summarizes an empirical evaluation of over-parameterized DeepONets with ReLU and smooth activations on a prototypical operator learning problem. In order to complement our theoretical results, we present empirical evaluation of our guarantees on three template operator learning problems: (i) Antiderivative operator, (ii) Diffusion-Reaction PDE, and (iii) Burger's equation and demonstrate that wider DeepONets lead to overall lower training loss at the end of the training process.

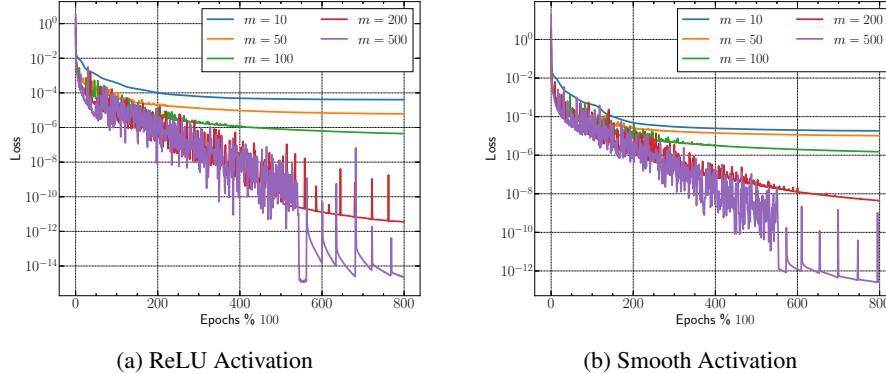

(a) ReLU Activation          (b) Smooth Activation

Figure 1: Benefits of over-parameterization on learning of DeepONets for the Antiderivative Operator: $G_{\boldsymbol{\theta}}(\boldsymbol{u})(x) = \int_0^x \boldsymbol{u}(\xi)\,\mathrm{d}\xi$. In both cases $m$ denotes the width of the branch net and the trunk net. For both ReLU and smooth activations, increasing the width $m$ leads to much lower losses. Note that the y-axis is in log-scale.

The rest of the paper is organized as follows. In Section 2 we review the existing literature on neural operators, operator networks and over-parameterization based approaches for establishing convergence guarantees for deep networks. Next, we devote Section 3 to briefly outline the the DeepONet model, the learning problem and the corresponding architecture. Section 4 contains the first technical result of the paper. In Section 4 we establish convergence guarantees for DeepONets with smooth activations (for both branch and trunk net) based on the Restricted Strong Convexity (RSC) of the loss. Next, in Section 5, we present the second technical result of the paper where we establish optimization guarantees for DeepONets with ReLU activations by showing that the Neural Tangent Kernel (NTK) of the DeepONet is positive definite at initialization. In Section 6 we present simple empirical evaluations of the main results by carrying out a parametric study based on increasing the DeepONet width and noting its effect on the total loss during training. We finally conclude by summarizing the main contributions in Section 7.

## 2 RELATED WORK

### 2.1 LEARNING OPERATOR NETWORKS

Constructing operator networks for ordinary differential equations (ODE's) using learning-based approaches was first studied in (Chen & Chen, 1995) where the authors showed that a neural network with a single hidden layer can approximate *a nonlinear continuous functional* to arbitrary accuracy. This was, in essence, akin to the Universal Approximation Theorem for classical neural networks (see, e.g., (Cybenko, 1989; Hornik et al., 1989; Hornik, 1991; Lu et al., 2017)). While the theorem only guaranteed the existence of a neural architecture, it was not practically realized until (Lu et al., 2021) which also provided an extension of the theorem to deep networks. Since then a number of works have pursued applications of DeepONets to different problems (see, e.g. (Goswami et al., 2022; Wang et al., 2021; Wang & Perdikaris, 2021)). Recently (Kontolati et al., 2022) studied the influence of over-parameterization on neural surrogates based on DeepONets in context of dynamical systems. While their paper studies the effects of over-parameterization on the generalization properties of DeepONets, an optimization analysis of DeepONets is a largely open problem.

### 2.2 OPTIMIZATION: NTK, ETC.

Optimization of over-parameterized deep networks have been studied extensively (see, e.g., (Du et al., 2019; Arora et al., 2019b;a; Allen-Zhu et al., 2019; Liu et al., 2021a)). In particular, (Jacot et al., 2018) showed that the neural tangent kernel (NTK) of a deep network converges to an explicit kernel in the limit of infinite network width and stays constant during training. (Liu et al., 2021a) showed that this constancy arises due to the scaling properties of the hessian of the predictor as a function of network width. (Du et al., 2019; Allen-Zhu et al., 2019) showed that gradient descent

converges to zero training error in polynomial time for a deep over-parameterized model, with (Du et al., 2019) showing it for a deep model with residual connections (ResNet) while (Allen-Zhu et al., 2019) showed in context of feed-forward models, CNNs and ResNets. (Karimi et al., 2016) showed that the Polyak-Lojasiewicz (PL) condition, a much weaker condition than strong convexity can be used to explain the linear convergence of gradient-based methods.

# 3  LEARNING DEEP OPERATOR NETWORKS

Learning neural operators (Li et al., 2020; Lu et al., 2021) is a fundamentally challenging problem as it requires learning parametric maps between two *infinite-dimensional* function spaces which is in sharp contrast to classical deep learning which learns parametric maps between two *finite-dimensional* vector spaces. A succinct review of learning in infinite dimensions is provided in Section A.1 in the Appendix. Here we focus on the DeepONet model and outline its main features.

## 3.1  DEEPONET SETUP

In what follows, we use the notation $\boldsymbol{f}(\boldsymbol{\theta}_f; \boldsymbol{u})$ to denote a deep network $\boldsymbol{f}_{\boldsymbol{\theta}_f} : \mathbb{R}^m \mapsto \mathbb{R}^n$ where $\boldsymbol{u}$ denotes the input and $\boldsymbol{\theta}_f$ the learnable parameters. A DeepONet is an operator network that learns a parametric map $G_{\boldsymbol{\theta}}$ such that $G_{\boldsymbol{\theta}}(u) \approx G^{\dagger}(u)$, where $u$ denotes the input function, and $G^{\dagger}$ denotes the "true" operator whose approximation we wish to learn. Following (Lu et al., 2021) a DeepONet predictor can defined as the inner product of two deep networks: $\boldsymbol{f} = \{f_k\}_{k=1}^K$ known as the branch net and $\boldsymbol{g} = \{g_k\}_{k=1}^K$ the trunk net, namely

$$G_{\boldsymbol{\theta}}(\boldsymbol{u})(\boldsymbol{y}) := \sum_{k=1}^K f_k(\boldsymbol{\theta}_f; \boldsymbol{u}) g_k(\boldsymbol{\theta}_g; \boldsymbol{y}), \tag{1}$$

where $\boldsymbol{u} \in \mathbb{R}^{d_u}$ is the input function and $y \in \mathrm{dom}(G_{\boldsymbol{\theta}}(\boldsymbol{u})) \subseteq \mathbb{R}^{d_v}$ the output locations[1]. The training data comprises of $n$ input functions, that is $\{u^{(i)}\}_{i=1}^n$ and $q_i$ output locations for each $G(u^{(i)})$, i.e. $\{\{y_j^{(i)}\}_{j=1}^{q_i}\}_{i=1}^n$ with $y_j^{(i)}$ denoting the $j$-th output location for $G_{\boldsymbol{\theta}}(u^{(i)})$. The branch net $\boldsymbol{f}$ has parameters $\boldsymbol{\theta}_f \in \mathbb{R}^{p_f}$ and the trunk net $\boldsymbol{g}$ has parameters $\boldsymbol{\theta}_g \in \mathbb{R}^{p_g}$. The entire set of parameters for the DeepONet is given by $\boldsymbol{\theta} = [\boldsymbol{\theta}_f^\top \ \boldsymbol{\theta}_g^\top]^\top \in \mathbb{R}^{p_f + p_g}$. Further, let $\{\boldsymbol{x}_r\}_{r=1}^R \in \mathrm{dom}(\boldsymbol{u}) \subseteq \mathbb{R}^d \, \forall r \in [R]$ and $u^{(i)}(x_r) \in \mathbb{R}^{d_u}$. For scalar functions $u^{(i)} \in \mathbb{R}$ the branch net takes input $\{u^{(i)}(x_r)\}_{r=1}^R$, which implies $\boldsymbol{f} : \mathbb{R}^R \mapsto \mathbb{R}^K$. Similarly, for scalar output locations $y_j^{(i)} \in \mathbb{R}$ we have $\boldsymbol{g} : \mathbb{R} \mapsto \mathbb{R}^K$. The DeepONet learning problem can then be cast as the minimization of the following empirical risk:

$$\boldsymbol{\theta}^{\dagger} = \operatorname*{arg\,min}_{\boldsymbol{\theta} \in \Theta} \mathcal{L}\left(G_{\boldsymbol{\theta}}(\boldsymbol{u}), G^{\dagger}(\boldsymbol{u})\right) = \frac{1}{n} \sum_{i=1}^n \frac{1}{q_i} \sum_{j=1}^{q_i} \ell\left(G_{\boldsymbol{\theta}}(u^{(i)})(y_j^{(i)}) - G^{\dagger}(u^{(i)})(y_j^{(i)})\right), \tag{2}$$

with

$$G_{\boldsymbol{\theta}}(u^{(i)})(y_j^{(i)}) = \sum_{k=1}^K f_k\left(\boldsymbol{\theta}_f; \{u^{(i)}(x_r)\}_{r=1}^R\right) g_k\left(\boldsymbol{\theta}_g; y_j^{(i)}\right), \tag{3}$$

and $\ell(z) := 1/2(z)^2$ denoting the mean-squared error (MSE) loss. Note that the "true" operator $G^{\dagger}$ whose approximation is sought in (2) can either be explicit, e.g. integral of a function, or implicit, e.g. the solution to a nonlinear partial differential equation (PDE).

## 3.2  DEEPONET ARCHITECTURE

We now briefly outline the architecture used throughout the analysis in Sections 4 and 5 and in the numerical experiments in Section 6. We adopt fully connected feedforward neural networks (FNNs) for both the branch and trunk nets which is also the baseline DeepONet model in (Lu et al., 2021). Figure 4 in the Appendix shows a schematic of the architecture and the notation used throughout this paper. For the architecture we adopt the unstacked configuration (see, Fig 1d in (Lu et al., 2021)).

---

[1]The original DeepONet paper puts forth the above model and another one with a bias term added to the inner product. For definiteness, we restrict our attention to the model without bias

**Remark 3.1** (DeepONet Training Tuple). As mentioned in the prequel, each DeepONet training data comprises of the tuple $\mathcal{D}^{(i)} := \left( \{u^{(i)}(x_r)\}_{r=1}^R, \{y_j^{(i)}\}_{j=1}^{q_i}, \{G(u^{(i)})(y_j^{(i)})\}_{j=1}^{q_i} \right)$. $\qquad\square$

**Remark 3.2** (Training Dataset). The entire training dataset $\mathcal{D}$ is then simply given by the collection of all tuples $\{\mathcal{D}^{(i)}\}_{i=1}^n$. $\qquad\square$

**Remark 3.3** (Widths $m_f$ and $m_g$). We denote the width of the branch net by $m_f$ and the trunk net by $m_g$ and use $m_f = m_g = m$ interchangeably through Sections 4-5. Similarly, for the experiments we use $m_f = m_g$ unless otherwise stated. For the latter (i.e. when $m_f \neq m_g$) the analysis in Section 4 still holds with $m = \min(m_f, m_g)$ $\qquad\square$

**Remark 3.4** (Last layer of the branch and trunk net). Note that the last layer of the branch and trunk net is simply a linear layer and does not have any nonlinearity/activation. $\qquad\square$

## 4 OPTIMIZATION GUARANTEES FOR DEEPONETS: SMOOTH ACTIVATIONS

In this section, we focus on DeepONets based on smooth activation functions. To build up to the optimization analysis, we first establish a bound on the spectral norm of the DeepONet predictor, in particular showing that $\|\nabla^2 G_{\boldsymbol{\theta}}(\boldsymbol{u})(\boldsymbol{y})\|_2 = O(\frac{1}{\sqrt{m}})$ where, again, $m = m_f = m_g$. The spectral norm bound is then used to establish a form of Restricted Strong Convexity (RSC) of the DeepONet loss (2), which in turn is used to establish geometric convergence of gradient descent (GD). For the analysis, analogous to (Liu et al., 2021b), we consider a FNN for the branch net:

$$\beta_f^{(0)} = \boldsymbol{u}, \quad \beta_f^{(l)} = \phi_l\left(\frac{1}{\sqrt{m_{\mathrm{f}}}}W_f^{(l)}\beta_f^{(l-1)}\right), \forall l \in [L-1], \quad \beta_f^{(L)} = W_f^{(L)}\beta_f^{(L-1)}$$
$$f_k = \beta_{f,k}^{(L)}, \quad \forall k \in [K]\,, \tag{4}$$

where $m_f = m$ and $L$ denote the width and depth of the branch net respectively, $[K] := \{1, \ldots, K\}$, $\phi_l$ is the activation function at layer $l$, $\beta_f^{(l)}$ are the outputs at layer $l$ and $W_f^{(l)} \equiv w_{f_{ij}}^{(l)}$ denote the weight matrices at layer $l$. Similarly, we consider a fully connected feedforward network for the trunk net:

$$\beta_g^{(0)} = \boldsymbol{y}, \quad \beta_g^{(l)} = \phi_l\left(\frac{1}{\sqrt{m_g}}W_g^{(l)}\beta_g^{(l-1)}\right), \forall l \in [L-1], \quad \beta_g^{(L)} = W_g^{(L)}\beta_g^{(L-1)}$$
$$g_k = \beta_{g,k}^{(L)}, \quad \forall k \in [K]\,, \tag{5}$$

where, again, $m_g = m$ and $L$ denote the width and depth of the trunk net respectively and $W_g^{(l)} \equiv w_{g_{ij}}^{(l)}$ denote the weight matrices at layer $l$ of the trunk net. In order to aid our analysis, we make the following assumptions on the activations, the loss and the weights:

**Assumption 1** (Activation functions). *The activation functions $\phi_l$ at each layer $l$ are 1-Lipschitz and $\beta_\phi$-smooth (i.e. $\phi'' \leq \beta_\phi$) for some $\beta_\phi > 0$.*

**Assumption 2** (Loss function). *We assume the loss $\ell_{i,j}$ is (i) strongly convex, i.e., $\ell''_{i,j} \geq a > 0$, (ii) smooth, i.e., $\ell''_{i,j} \leq b$, and (iii) Lipschitz, i.e., $\left|\ell'_{i,j}\right| \leq \lambda$., where we make use of the following notation*

$$\ell_{i,j} \equiv \ell\left(G_{\boldsymbol{\theta}}(u^{(i)})(y_j^{(i)}) - G^\dagger(u^{(i)})(y_j^{(i)})\right), \quad \ell'_{i,j} \equiv \ell'\left(G_{\boldsymbol{\theta}}(u^{(i)})(y_j^{(i)}) - G^\dagger(u^{(i)})(y_j^{(i)})\right),$$
$$\ell''_{i,j} \equiv \ell''\left(G_{\boldsymbol{\theta}}(u^{(i)})(y_j^{(i)}) - G^\dagger(u^{(i)})(y_j^{(i)})\right). \tag{6}$$

**Assumption 3** (Initialization of Weights). *All weights of the branch and trunk net are initialized as $w_{f_{ij}}^{(l)}|_{t=0} = w_{f_{0,\,ij}}^{(l)} \sim \mathcal{N}(0, \sigma_0^2)$ and $w_{g_{ij}}^{(l)}|_{t=0} = w_{g_{0,\,ij}}^{(l)} \sim \mathcal{N}(0, \sigma_0^2)$ for $l \in [L-1]$ and some constant $\sigma_0 > 0$ respectively. Furthermore, in order for the model (1) to have a suitable scaling in our analysis, we initialize the weights in the last layer of the branch and trunk nets as $w_{f_{0,\,ij}}^L \sim \mathcal{N}(0, 1/(m\,K))$ and $w_{g_{0,\,ij}}^L \sim \mathcal{N}(0, 1/(m\,K))$ respectively.*

**Remark 4.1.** For Assumption 1, our analysis straightforwardly extends to general $\varsigma$-Lipschitz activations, with constants depending $\varsigma$. For Assumption 2, the Lipschitz loss assumption can be dropped by assuming that the true responses $G^\dagger(\boldsymbol{u})(\boldsymbol{y})$ are bounded and showing that the prediction responses $G_{\boldsymbol{\theta}}(\boldsymbol{u})(\boldsymbol{y})$ are bounded with high probability. $\qquad\square$

**Definition 1** (Norm ball). *Our analysis focuses on the standard Euclidean norm ball around the initialization $\boldsymbol{\theta}_0$, i.e. $B_\rho^{\mathrm{Euc}}(\boldsymbol{\theta}_0)$, where*

$$B_\rho^{\mathrm{Euc}}(\bar{\boldsymbol{\theta}}) := \left\{ \boldsymbol{\theta} \in \mathbb{R}^{p_f+p_g} \mid \left\| \boldsymbol{\theta} - \bar{\boldsymbol{\theta}} \right\|_2 \leq \rho \right\}. \tag{7}$$

## 4.1 Spectral Norm of the Hessian of branch and trunk Nets

The convergence analysis makes use of the gradients and hessians of the total loss (2) and the predictor (1) with respect to the parameters $\boldsymbol{\theta}$, namely,

$$\nabla_{\boldsymbol{\theta}} \mathcal{L}(\boldsymbol{\theta}) = \left[ \nabla_{\boldsymbol{\theta}_f} \mathcal{L}; \nabla_{\boldsymbol{\theta}_g} \mathcal{L} \right], \quad \text{and} \quad \nabla_{\boldsymbol{\theta}}^2 \mathcal{L} = \boldsymbol{H}(\boldsymbol{\theta}) = \left[ \begin{array}{cc} H_{ff} & H_{fg} \\ H_{gf} & H_{gg} \end{array} \right], \tag{8}$$

where $\nabla_{\boldsymbol{\theta}_f} \mathcal{L}(\boldsymbol{\theta}) \in \mathbb{R}^{p_f}$ and $\nabla_{\boldsymbol{\theta}_g} \mathcal{L}(\boldsymbol{\theta}) \in \mathbb{R}^{p_g}$. Note that we make use of the notation $\nabla_{\boldsymbol{\theta}_f}(\cdot)$ to denote the derivative wrt the parameters $\boldsymbol{\theta}_f$ and this *is not* a functional gradient. Similarly, the individual blocks in the $2 \times 2$ block hessian $\boldsymbol{H}(\boldsymbol{\theta})$ are given by

$$H_{ff} = \frac{\partial^2 \mathcal{L}}{\partial \boldsymbol{\theta}_f^2}, \quad H_{fg} = \frac{\partial^2 \mathcal{L}}{\partial \boldsymbol{\theta}_f \partial \boldsymbol{\theta}_g}, \quad H_{gf} = H_{fg}^\top = \frac{\partial^2 \mathcal{L}}{\partial \boldsymbol{\theta}_g \partial \boldsymbol{\theta}_f}, \quad \text{and} \quad H_{gg} = \frac{\partial^2 \mathcal{L}}{\partial \boldsymbol{\theta}_g^2}, \tag{9}$$

where $H_{ff} \in \mathbb{R}^{p_f \times p_f}$, $H_{gg} \in \mathbb{R}^{p_g \times p_g}$, $H_{fg} \in \mathbb{R}^{p_f \times p_g}$, $H_{gf} \in \mathbb{R}^{p_g \times p_f}$ and the argument $\boldsymbol{\theta}$ is ignored for clarity of exposition. Using (2) and rewriting the derivatives in (8) and (9), we get

$$\frac{\partial \mathcal{L}}{\partial \boldsymbol{\theta}_f} = \frac{1}{n} \sum_{i=1}^n \frac{1}{q_i} \sum_{j=1}^{q_i} \ell_{i,j}' \sum_{k=1}^K g_{k,j}^{(i)} \nabla_{\boldsymbol{\theta}_f} f_k^{(i)} \quad \text{and} \quad \frac{\partial \mathcal{L}}{\partial \boldsymbol{\theta}_g} = \frac{1}{n} \sum_{i=1}^n \frac{1}{q_i} \sum_{j=1}^{q_i} \ell_{ij}' \sum_{k=1}^K f_k^{(i)} \nabla_{\boldsymbol{\theta}_g} g_{k,j}^{(i)}, \tag{10}$$

for the gradients, and

$$\frac{\partial^2 \mathcal{L}}{\partial \boldsymbol{\theta}_f^2} = \frac{1}{n} \sum_{i=1}^n \frac{1}{q_i} \sum_{j=1}^{q_i} \ell_{i,j}' \sum_{k=1}^K g_{k,j}^{(i)} \nabla_{\boldsymbol{\theta}_f}^2 f_k^{(i)} + \frac{1}{n} \sum_{i=1}^n \frac{1}{q_i} \sum_{j=1}^{q_i} \ell_{i,j}'' \left( \sum_{k,\hat{k}=1}^K g_{k,j}^{(i)} g_{\hat{k},j}^{(i)} \nabla_{\boldsymbol{\theta}_f} f_k^{(i)} \nabla_{\boldsymbol{\theta}_f} f_{\hat{k}}^{(i)\top} \right),$$

$$\frac{\partial^2 \mathcal{L}}{\partial \boldsymbol{\theta}_g^2} = \frac{1}{n} \sum_{i=1}^n \frac{1}{q_i} \sum_{j=1}^{q_i} \ell_{i,j}' \sum_{k=1}^k f_k^{(i)} \nabla_{\boldsymbol{\theta}_g}^2 g_{k,j}^{(i)} + \frac{1}{n} \sum_{i=1}^n \frac{1}{q_i} \sum_{j=1}^{q_i} \ell_{i,j}'' \left( \sum_{k,\hat{k}=1}^K f_k^{(i)} f_{\hat{k}}^{(i)} \nabla_{\boldsymbol{\theta}_g} g_{k,j}^{(i)} \nabla_{\boldsymbol{\theta}_g} g_{\hat{k},j}^{(i)\top} \right),$$

$$\frac{\partial^2 \mathcal{L}}{\partial \boldsymbol{\theta}_f \partial \boldsymbol{\theta}_g} = \underbrace{\frac{1}{n} \sum_{i=1}^n \frac{1}{q_i} \sum_{j=1}^{q_i} \ell_{i,j}' \sum_{k=1}^K \nabla_{\boldsymbol{\theta}_f} f_k^{(i)} \nabla_{\boldsymbol{\theta}_g} g_{k,j}^{(i)\top}}_{=H_{fg}^{(1)}} + \underbrace{\frac{1}{n} \sum_{i=1}^n \frac{1}{q_i} \sum_{j=1}^{q_i} \ell_{i,j}'' \left( \sum_{k,\hat{k}=1}^K g_{k,j}^{(i)} f_{\hat{k}}^{(i)} \nabla_{\boldsymbol{\theta}_f} f_k^{(i)} \nabla_{\boldsymbol{\theta}_g} g_{\hat{k},j}^{(i)\top} \right)}_{=H_{fg}^{(2)}},$$

$$\tag{11}$$

for the individual blocks of the hessian (8) where, we make use of the notation $g_{k,j}^{(i)} = g_k(\boldsymbol{\theta}_g; y_j^{(i)})$ and $f_k^{(i)} = f_k(\boldsymbol{\theta}_f; u^{(i)})$.

**Lemma 4.1.** Under Assumptions 1, 2 and 3, and for $\boldsymbol{\theta} \in B_\rho^{\mathrm{Euc}}(\boldsymbol{\theta}_0)$, with high-probability we have for all $k \in [K]$

$$\max_{i \in [n]} \left\| \nabla_{\boldsymbol{\theta}_f}^2 f_k^{(i)} \right\| \leq \frac{c^{(f)}}{\sqrt{m_f}}, \quad \text{and} \quad \max_{i \in [n]} \max_{j \in [q_i]} \left\| \nabla_{\boldsymbol{\theta}_g}^2 g_{k,j}^{(i)} \right\| \leq \frac{c^{(g)}}{\sqrt{m_g}} \tag{12}$$

$$\left\| \nabla_{\boldsymbol{\theta}_f} f_k \right\|_2 \leq \varrho^{(f)}, \quad \text{and} \quad \left\| \nabla_{\boldsymbol{\theta}_g} g_k \right\| \leq \varrho^{(g)}, \tag{13}$$

where $c^{(f)}$, $c^{(g)}$, $\varrho^{(f)}$, $\varrho^{(g)}$ are suitable constants, $\nabla_{\boldsymbol{\theta}_f}(\cdot) = \partial(\cdot)/\partial \boldsymbol{\theta}_f$, $\nabla_{\boldsymbol{\theta}_g}(\cdot) = \partial(\cdot)/\partial \boldsymbol{\theta}_g$, $\nabla_{\boldsymbol{\theta}_f}^2(\cdot) = \partial^2(\cdot)/\partial \boldsymbol{\theta}_f^2$ and $\nabla_{\boldsymbol{\theta}_g}^2(\cdot) = \partial^2(\cdot)/\partial \boldsymbol{\theta}_g^2$.

*Proof.* The proof follows directly from Theorem 3.2 in (Liu et al., 2021a). □

## 4.2 GEOMETRIC CONVERGENCE BASED ON RESTRICTED STRONG CONVEXITY

We now focus on establishing the convergence of gradient descent (GD) by using the Hessian spectral norm bound. The convergence analysis is based on a generalization of the notion of restricted strong convexity (RSC) of the loss (2) (see (Negahban et al., 2012; Negahban & Wainwright, 2012; Zhang & Cheng, 2015; Zhang & Yin, 2013; Lai & Yin, 2013) for a review of RSC and its applications in high-dimensional statistics for linear models). In order to further aid clarity of exposition, we state the main results along with their implications in this section and leave the details of the proofs to Section A.3 in the Appendix.

**Definition 2** (Restricted Strong Convexity (RSC)). *A function $\mathcal{L}$ is said to satisfy $\alpha$-restricted strong convexity ($\alpha$-RSC) with respect to the tuple $(Q, \boldsymbol{\theta})$ if for any $\boldsymbol{\theta}' \in Q \subseteq \mathbb{R}^p$ and some fixed $\boldsymbol{\theta} \in \mathbb{R}^p$, we have $\mathcal{L}(\boldsymbol{\theta}') \geq \mathcal{L}(\boldsymbol{\theta}) + \langle \boldsymbol{\theta}' - \boldsymbol{\theta}, \nabla_{\boldsymbol{\theta}} \mathcal{L}(\boldsymbol{\theta}) \rangle + \frac{\alpha}{2} \|\boldsymbol{\theta}' - \boldsymbol{\theta}\|_2^2$, with $\alpha > 0$.*

Note that $\mathcal{L}$ being $\alpha$-RSC w.r.t. $(S, \boldsymbol{\theta})$ does not need $\mathcal{L}$ to be convex on $\mathbb{R}^p$. Further, let $\{\boldsymbol{\theta}_t\}_{t \geq 0}$ denote a sequence of iterates obtained from GD, i.e.,

$$\boldsymbol{\theta}_{t+1} = \boldsymbol{\theta}_t - \eta_t \nabla_{\boldsymbol{\theta}} \mathcal{L}(\boldsymbol{\theta}_t) \ . \tag{14}$$

Since $\boldsymbol{\theta}_t^\top = [\boldsymbol{\theta}_{f,t}^\top \ \boldsymbol{\theta}_{g,t}^\top]^\top$, we will use dynamic restricted sets $Q_\kappa^t \subseteq \mathbb{R}^p$, where $p = p_f + p_g$, to show RSC of the DeepONet loss. Note that these sets are parameterized by a constant $\kappa$ which measures the absolute cosine similarity between suitable vectors (see Section A.3 and specifically Proposition 1 in the Appendix for a detailed discussion on these sets). Further, our optimization for DeepONets is based on a second order Taylor expansion of the loss (2) w.r.t. $\boldsymbol{\theta}_t = [\boldsymbol{\theta}_{f,t}^\top \ \boldsymbol{\theta}_{g,t}^\top]^\top$:

$$\mathcal{L}(\boldsymbol{\theta}) = \mathcal{L}(\boldsymbol{\theta}_t) + \langle \boldsymbol{\theta} - \boldsymbol{\theta}_t, \nabla_{\boldsymbol{\theta}} \mathcal{L}(\boldsymbol{\theta}_t) \rangle + \frac{1}{2}(\boldsymbol{\theta} - \boldsymbol{\theta}_t)^\top \boldsymbol{H}(\tilde{\boldsymbol{\theta}})(\boldsymbol{\theta} - \boldsymbol{\theta}_t) \ , \tag{15}$$

where, $\nabla_{\boldsymbol{\theta}} \mathcal{L}(\boldsymbol{\theta}_t)$ and $\boldsymbol{H}(\tilde{\boldsymbol{\theta}})$ are as in (8), (9), (10), and (11), and $\tilde{\boldsymbol{\theta}} = \tau \boldsymbol{\theta} + (1 - \tau)\bar{\boldsymbol{\theta}}$ for some $\tau \in [0, 1]$.

**Definition 3** ($Q_\kappa^t$ sets.). *For an iterate $\boldsymbol{\theta}_t = [\boldsymbol{\theta}_{f,t}^\top \ \boldsymbol{\theta}_{g,t}^\top]^\top$, consider the singular value decomposition*

$$\frac{1}{n} \sum_{i=1}^n \frac{1}{q_i} \sum_{j=1}^{q_i} \ell'_{i,j} \sum_{k=1}^K \nabla_{\boldsymbol{\theta}_f} f_k^{(i)} \nabla_{\boldsymbol{\theta}_g} g_{k,j}^{(i)\,\top} = \sum_{h=1}^{\tilde{q}} \sigma_h \boldsymbol{a}_h \boldsymbol{b}_h^\top \ , \tag{16}$$

*where $\tilde{q} \leq qk$, and $\sigma_h > 0, \boldsymbol{a}_h \in \mathbb{R}^{p_f}, \boldsymbol{b}_h \in \mathbb{R}^{p_g}$ respectively denote the singular values, left singular vectors, and right singular vectors. Further, let*

$$\bar{G}_{\boldsymbol{\theta}} = \frac{1}{n} \sum_{i=1}^n \frac{1}{q_i} \sum_{j=1}^{q_i} G_{\boldsymbol{\theta}}(u^{(i)})(y_j^{(i)}) \ . \tag{17}$$

*Then, for some suitable $\kappa \in (0, \frac{1}{2}]$, we define the set:*

$$Q_\kappa^t := \left\{ \boldsymbol{\theta}' = [\boldsymbol{\theta}_f'; \boldsymbol{\theta}_g'] : |\cos(\boldsymbol{\theta}' - \boldsymbol{\theta}_t, \nabla_{\boldsymbol{\theta}} \bar{G}_{\boldsymbol{\theta}_t})| \geq \kappa, \right.$$
$$\left. \sum_{h=1}^{\tilde{q}} \sigma_h \langle \boldsymbol{\theta}_f' - \boldsymbol{\theta}_{f,t}, \boldsymbol{a}_h \rangle \langle \boldsymbol{\theta}_g' - \boldsymbol{\theta}_{g,t}, \boldsymbol{b}_h \rangle \geq 0 \ , \ \forall h \in [\tilde{q}] \right\} \ . \tag{18}$$

**Remark 4.2.** Note that $p_f, p_g$ are respectively the number of parameters in the branch and trunk nets and the models can be sufficiently over-parameterized such that they are larger than the number of training examples $q = \sum_{i=1}^n q_i$. □

**Remark 4.3** ($Q_\kappa^t$ sets). The specifics of the restricted set $Q_\kappa^t \in \mathbb{R}^{p_f+p_g}$ stem from technicalities in the analysis. For a detailed outline of these sets we refer the reader to Section A.3 and specifically Proposition 1 in the Appendix. □

**Theorem 4.2** (RSC of the loss). Under the assumptions Assumptions 1, 2 and 3, $\forall \boldsymbol{\theta}' \in B_\kappa^t := Q_\kappa^t \cap B_\rho^{\text{Euc}}(\boldsymbol{\theta}_0)$ with high probability we have

$$\mathcal{L}(\boldsymbol{\theta}') \geq \mathcal{L}(\boldsymbol{\theta}_t) + \langle \boldsymbol{\theta}' - \boldsymbol{\theta}_t, \nabla_{\boldsymbol{\theta}} \mathcal{L}(\boldsymbol{\theta}_t) \rangle + \frac{\alpha_t}{2} \|\boldsymbol{\theta}' - \boldsymbol{\theta}_t\|_2^2, \quad \text{where} \quad \alpha_t = c_1 \|\nabla_{\boldsymbol{\theta}} \bar{G}_t\|_2^2 - \frac{c_2}{\sqrt{m}},$$
(19)

where $\bar{G}_t = \frac{1}{n} \sum_{i=1}^n \frac{1}{q_i} \sum_{j=1}^{q_i} G_{\boldsymbol{\theta}_t}(u^{(i)})(y_j^{(i)})$ and $c_1, c_2 > 0$ are constants. Thus, the loss $\mathcal{L}(\boldsymbol{\theta})$ satisfies RSC w.r.t $(B_\kappa^t, \boldsymbol{\theta}_t)$ whenever $\alpha_t > 0$.

*Proof.* A detailed proof is presented in Section A.3 in the Appendix. ☐

**Theorem 4.3** (Smoothness of Loss). Under the assumptions Assumptions 1, 2 and 3, with high probability, for $\boldsymbol{\theta} \in B_\rho^{\text{Euc}}(\boldsymbol{\theta}_0)$, $\mathcal{L}(\boldsymbol{\theta})$ is $\beta$-smooth with $\beta = b\varrho^2 + \frac{c\sqrt{\lambda}}{\sqrt{m}}$ with $c = \max(c^{(f)}, c^{(g)})$, $\varrho = \max(\varrho^{(f)}, \varrho^{(g)})$ with $c^{(f)}, c^{(g)}, \varrho^{(f)}, \varrho^{(g)}$ as in Lemma 4.1.

*Proof.* We again refer the reader to Section A.3 in the Appendix for a detailed proof. ☐

**Lemma 4.4** (RSC $\implies$ Restricted PL). The RSC and Smoothness of the Loss together imply a form of Polyak-Łojasiewicz (PL) condition w.r.t. the tuple $(B_t, \boldsymbol{\theta}_t)$, unlike standard PL which holds without restrictions (Karimi et al., 2016).

*Proof.* For a detailed outline of the proof, we refer the reader to Section A.3 in the Appendix. ☐

**Theorem 4.5** (Global Loss Reduction). Consider the same conditions as in Theorem 4.3 and $\alpha_t > 0$ for $t \in [T]$ for a gradient descent update $\boldsymbol{\theta}_{t+1} = \boldsymbol{\theta}_t - \eta_t \nabla_{\boldsymbol{\theta}} \mathcal{L}(\boldsymbol{\theta}_t)$ with $\eta_t = \frac{\omega_t}{\beta}$ for some $\omega_t \in (0, 2)$, where $\beta$ is defined as in Theorem 4.3. Then, with high probability, $\forall \bar{\boldsymbol{\theta}} \in \underset{\boldsymbol{\theta} \in B_\rho^{\text{Euc}}(\boldsymbol{\theta}_0)}{\arginf} \mathcal{L}(\boldsymbol{\theta})$ with $0 \leq \gamma_t := \frac{\mathcal{L}(\bar{\boldsymbol{\theta}}_{t+1}) - \mathcal{L}(\bar{\boldsymbol{\theta}})}{\mathcal{L}(\boldsymbol{\theta}_t) - \mathcal{L}(\bar{\boldsymbol{\theta}})} < 1$ and $\bar{\boldsymbol{\theta}}_{t+1} \in \arginf_{\boldsymbol{\theta} \in Q_\kappa^t \cap B_\rho^{\text{Euc}}(\boldsymbol{\theta}_0)} \mathcal{L}(\boldsymbol{\theta})$, we have

$$\mathcal{L}(\boldsymbol{\theta}_{t+1}) - \mathcal{L}(\bar{\boldsymbol{\theta}}) \leq \left( 1 - \frac{\alpha_t \omega_t (1 - \gamma_t)}{\beta}(2 - \omega_t) \right) (\mathcal{L}(\boldsymbol{\theta}_t) - \mathcal{L}(\bar{\boldsymbol{\theta}})).$$
(20)

*Proof.* We refer the reader to Section A.3 and in particular Lemma 1 in the Appendix for the proof. ☐

**Remark 4.4.** A direct consequence of Theorem 4.5 is global reduction in the loss in $\mathbb{R}^p$ and thus the convergence of gradient descent for the DeepONet optimization problem. ☐

## 5 OPTIMIZATION GUARANTEES FOR DEEPONETS: RELU ACTIVATIONS

We now present an alternative optimization analysis[2], based on the Neural Tangent Kernel (NTK) (Jacot et al., 2018), to establish guarantees for the convergence of gradient descent and its variants for DeepONets. Although the NTK convergence theory holds for both smooth and ReLU activations (see, e.g. (Allen-Zhu et al., 2019; Du et al., 2019)), we present this in context of the latter. Further, in this work we only present a proof for the positive definiteness of the DeepONet NTK at initialization. This allows one to readily transcribe analogous existing arguments for deep networks, for the convergence of gradient descent, to the DeepONet model. (Allen-Zhu et al., 2019; Du et al., 2019; Nguyen et al., 2021) Now, recall the DeepONet predictor

$$G_{\boldsymbol{\theta}}\left(u^{(i)}\right)\left(y_j^{(i)}\right) := \sum_{k=1}^K f_k\left(\boldsymbol{\theta}_f; u^{(i)}\right) g_k\left(\boldsymbol{\theta}_g; y_j^{(i)}\right),$$
(21)

---

[2]The hessian-based analysis presented in Section 4 constitutes a sufficient condition and is not directly applicable for the case of ReLU activations. In contrast, the NTK analysis presented here applies to both smooth and ReLU activations.

and its corresponding gradient with respect to the parameters $\boldsymbol{\theta}$,

$$\nabla_{\boldsymbol{\theta}} G_{\boldsymbol{\theta}}(u^{(i)})(y_j^{(i)}) = \sum_{k=1}^{K} \begin{bmatrix} g_k\left(\boldsymbol{\theta}_g; y_j^{(i)}\right) \nabla_{\boldsymbol{\theta}_f} f_k\left(\boldsymbol{\theta}_f; u^{(i)}\right) \\ f_k\left(\boldsymbol{\theta}_f; u^{(i)}\right) \nabla_{\boldsymbol{\theta}_g} g_k\left(\boldsymbol{\theta}_g; y_j^{(i)}\right) \end{bmatrix}. \tag{22}$$

This allows us to write the NTK $\mathcal{K}(\boldsymbol{\theta})$ of the DeepONet, which is a $q \times q$ matrix as:

$$\mathcal{K}(\boldsymbol{\theta}) = \left[\left\langle \nabla_{\boldsymbol{\theta}} G_{\boldsymbol{\theta}}(u^{(i)})(y_j^{(i)}), \nabla_{\boldsymbol{\theta}} G_{\boldsymbol{\theta}}(u^{(i')})(y_{j'}^{(i')})\right\rangle\right]_{q \times q}, \tag{23}$$

where, again, $q = \sum_{i=1}^{n} q_i$. Given the branch and trunk nets are initialized using standard initialization techniques, as typically done in practice for deep networks, (Arora et al., 2019b; Du et al., 2019) with the addition of Assumption 3, the resulting branch net NTK $\mathcal{K}_{f,k}$ and trunk net NTK $\mathcal{K}_{g,k}$, namely

$$\begin{aligned} \mathcal{K}_{f,k} &= \left[\left\langle \nabla_{\boldsymbol{\theta}_{f,k}} f_k\left(\boldsymbol{\theta}_f; u^{(i)}\right), \nabla_{\boldsymbol{\theta}_{f,k}} f_k\left(\boldsymbol{\theta}_f; u^{(i')}\right)\right\rangle\right]_{n \times n} \\ \mathcal{K}_{g,k} &= \left[\left\langle \nabla_{\boldsymbol{\theta}_{g,k}} g_k\left(\boldsymbol{\theta}_g; y_j^{(i)}\right), \nabla_{\boldsymbol{\theta}_{g,k}} g_k\left(\boldsymbol{\theta}_g; y_{j'}^{(i')}\right)\right\rangle\right]_{q \times q}, \end{aligned} \tag{24}$$

can be shown to be positive definite with high probability for each $k \in [K]$ (Nguyen, 2021; Liu et al., 2021a; Du et al., 2019). In particular, with high probability, for $k \in [K]$, we have

$$\lambda_{\min}(\mathcal{K}_{f,k}) \geq \lambda_{0,f}, \qquad \text{and} \qquad \lambda_{\min}(\mathcal{K}_{g,k}) \geq \lambda_{0,g} \quad \forall\, k \in [K], \tag{25}$$

where $\lambda_{\min}(\mathcal{K}_{f,k})$ and $\lambda_{\min}(\mathcal{K}_{g,k})$ are the minimum eigenvalues of the branch and trunk net NTKs respectively, and $\lambda_{0,f}, \lambda_{0,g} > 0$ are positive constants (see, e.g., Theorem 4.1 in (Nguyen, 2021)).

**Theorem 5.1** ($\mathcal{K}(\boldsymbol{\theta})$ is positive definite at initialization). Given standard initialization for the branch and trunk nets, and granted that the individual branch and trunk net NTKs are positive definite (24)-(25), the NTK of DeepONet is positive definite at initialization, i.e.

$$\boldsymbol{\alpha}^{\top} \mathbb{E}[\mathcal{K}(\boldsymbol{\theta})] \boldsymbol{\alpha} \geq c > 0, \tag{26}$$

where $\boldsymbol{\alpha}$ denotes an arbitrary block unit vector with $n$ blocks, and $\alpha_{i,j}$ corresponds to the $j$-th entry in the $i$-th block and $c$ denotes a positive constant.

*Proof.* We refer the reader to Section A.4 and specifically Propositions 2, 3 and 4  □

**Remark 5.1.** A direct consequence of Theorem 5.1 is that the DeepONet NTK is positive definite at initialization. It is then straightforward to invoke standard NTK analysis to show convergence of gradient descent during training, see, e.g. (Jacot et al., 2018; Du et al., 2019; Arora et al., 2019b;a; Allen-Zhu et al., 2019; Nguyen et al., 2021).  □

## 6 EXPERIMENTS

We now turn to an empirical evaluation of the effect of over-parameterization on the training performance of DeepONets, as measured by the empirical risk over a mini-batch $B$ of the training dataset (3.2), for three canonical operator-learning problems. The results for smooth activations empirically verify the analysis in Section 4 whereas the ones for ReLU activations verify the analysis in Section 5. In all the examples described below, we consider FNN architectures for both branch and trunk nets which are similar to the ones chosen in (Lu et al., 2021). For definiteness, we set the width in each layer of the branch and trunk net to be the same (i.e. $m_f = m_g = m$) and then increase it uniformly from $m = 10$ to $m = 500$. We monitor the training process over $80,000$ training epochs and report the resulting average loss over each mini-batch with size $n_B$,

$$\mathcal{L}_{\mathcal{D}_B} := \frac{1}{n_B} \sum_{i=1}^{n_B} \frac{1}{q_i} \sum_{j=1}^{q_i} \ell\left(G_{\boldsymbol{\theta}}(u^{(i)})(y_j^{(i)}) - G^{\dagger}(u^{(i)})(y_j^{(i)})\right), \tag{27}$$

where $n_B$ denotes the number of input training functions $(u^i)$ in the batch $B$. We refer the reader to Section A.5 in the Appendix for specific details of the setup. Figure 2 shows the training loss (27) as a function of the epochs for DeepONets with smooth activations and Figure 3 shows the same for ReLU activations.

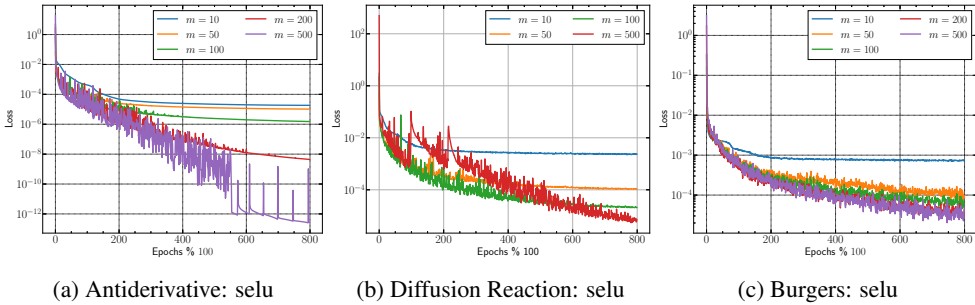

|     |     |     |
| --- | --- | --- |
| (a) Antiderivative: selu | (b) Diffusion Reaction: selu | (c) Burgers: selu |

Figure 2: Training progress of DeepONet with a smooth activation function selu (Klambauer et al., 2017) for (a) Antiderivative Operator, (b) Diffusion-Reaction Equation and (c) Burger's Equation. We plot the training loss (27) as a function of the epochs with the y-axis on a log-scale to clearly distinguish the effect of increasing width ($m$). Increasing the width $m$ of the branch and trunk net leads to lower losses for all the problems.

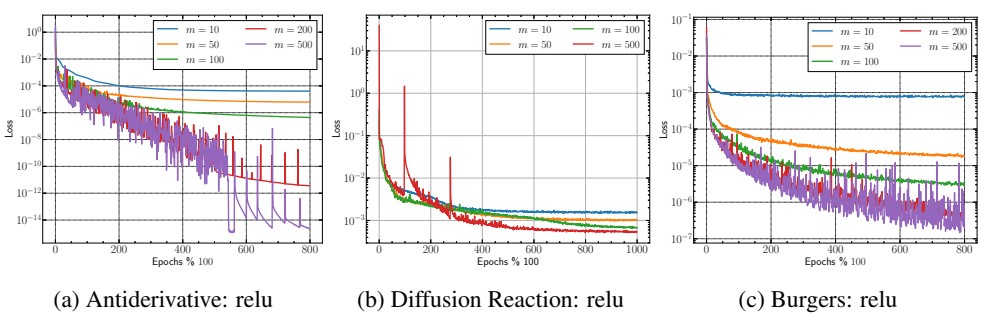

|     |     |     |
| --- | --- | --- |
| (a) Antiderivative: relu | (b) Diffusion Reaction: relu | (c) Burgers: relu |

Figure 3: Training progress of DeepONet with ReLU activations for (a) Antiderivative Operator, (b) Diffusion-Reaction Equation and (c) Burger's Equation. The y-axis is again plotted on a log-scale to clearly demarcate the effect of increasing width. Increasing the width $m$ again leads to lower training losses.

**Remark 6.1.** We store the mini-batch training loss at every 100-th training epoch and observe that the training loss measured over the mini-batch is lower for wider DeepONets. This observation is consistent for both smooth (selu) and non-smooth (relu) activations. □

**Remark 6.2** (Antiderivative Operator). The Antiderivative operator is a linear operator and hence is learned very accurately especially for wider DeepONets ($\mathcal{L}_{\mathcal{D}_B} \sim 10^{-12}$ at the end of training). □

**Remark 6.3** (Diffusion Reaction). The Diffusion reaction equation also demonstrates lower loss with increasing width, albeit less markedly than the antiderivative operator. This can be attributed in part to the fact that the operator is inherently nonlinear. □

**Remark 6.4** (Burger's equation). The operator corresponding to Burger's equation is more intricate with the added periodicity constraints on the solution (see Section A.5.3 in the appendix for details on the problem). We remark the distinction from the operator learning problem for Burger's equation studied in (Li et al., 2021a) where the operator only sought to learn the mapping from the input (initial condition $t = 0$) to the final output $t = 1$ and not the entire solution space $(x, t) \in [0, 1] \times [0, 1]$. □

## 7 DISCUSSION AND CONCLUSION

We present two novel optimization analyses for DeepONets (Lu et al., 2021) based on over-parameterization and establish convergence guarantees for the DeepONet models with smooth and ReLU activations. The analysis for smooth activations is built on top of the restricted strong convexity of the DeepONet loss whereas the one for ReLU activations is based on the positive definiteness of the NTK at initialization. To the best of our knowledge, this is the first work to mathematically and empirically show the benefits of over-parameterization on the the learning of DeepONets.

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

# A    APPENDIX

## A.1    LEARNING OPERATORS

Here we briefly outline the notion of learning for neural operators (Li et al., 2021a; 2020; Lu et al., 2021). The standard operator learning problem seeks to approximate a possibly nonlinear operator $G^\dagger : \mathcal{U} \mapsto \mathcal{V}$ by a parametric operator $G_{\boldsymbol{\theta} \in \Theta} : \mathcal{U} \mapsto \mathcal{V}$ that depends on the learnable parameters $\boldsymbol{\theta}$. The goal is to learn an optimal set of parameters $\boldsymbol{\theta}^\dagger$ such that $G_{\theta^\dagger} \approx G^\dagger$. Given observations $\{u^{(j)}\}_{j=1}^n \in \mathcal{U}$ and $\{G^\dagger(u^{(j)})\}_{j=1}^n \in \mathcal{V}$ where $u^{(j)} \sim \mu$ is an i.i.d sequence from the probability measure $\mu$ supported on $\mathcal{U}$ and $G(u^{(j)})$ is possibly corrupted with noise, the objective is to find $\theta^\dagger$ as the solution of the minimization problem

$$\boldsymbol{\theta}^\dagger = \underset{\boldsymbol{\theta} \in \Theta}{\arg\min} \, \mathbb{E}_{\boldsymbol{u} \sim \mu} \left[ \mathcal{C} \left( G_{\boldsymbol{\theta}}(\boldsymbol{u}), G^\dagger(\boldsymbol{u}) \right) \right], \tag{28}$$

where $\mathcal{U}$ and $\mathcal{V}$ are separable Banach spaces and $\mathcal{C}$ a suitable cost functional. This is analogous to the notion of learning in finite dimensions, which is precisely the setup classical deep learning used for.

## A.2    DEEPONET ARCHITECTURE

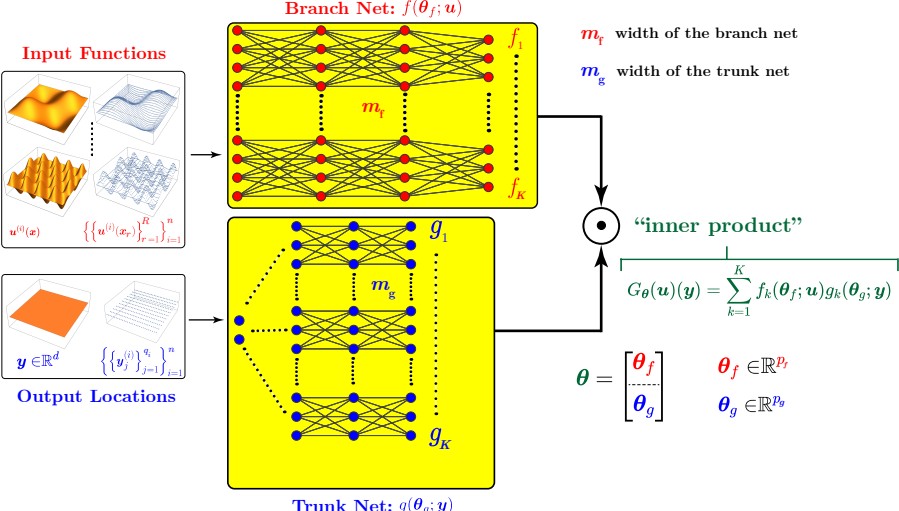

Figure 4: A schematic of the *unstacked* DeepONet architecture (Lu et al., 2021) used in this study. Note that the input functions need not be sampled on a structured grid of points in general.

## A.3    OPTIMIZATION GUARANTEES FOR DEEPONETS: SMOOTH ACTIVATIONS

**Theorem 4.2** (RSC of the loss). Under the assumptions Assumptions 1, 2 and 3, $\forall \boldsymbol{\theta}' \in B_\kappa^t :=$ $Q_\kappa^t \cap B_\rho^{\mathrm{Euc}}(\boldsymbol{\theta}_0)$ with high probability we have

$$\mathcal{L}\left(\boldsymbol{\theta}'\right) \geq \mathcal{L}(\boldsymbol{\theta}_t) + \langle \boldsymbol{\theta}' - \boldsymbol{\theta}_t, \nabla_{\boldsymbol{\theta}} \mathcal{L}(\boldsymbol{\theta}_t) \rangle + \frac{\alpha_t}{2} \|\boldsymbol{\theta}' - \boldsymbol{\theta}_t\|_2^2, \quad \text{where} \quad \alpha_t = c_1 \|\nabla_{\boldsymbol{\theta}} \bar{G}_t\|_2^2 - \frac{c_2}{\sqrt{m}}, \tag{19}$$

where $\bar{G}_t = \frac{1}{n} \sum_{i=1}^n \frac{1}{q_i} \sum_{j=1}^{q_i} G_{\boldsymbol{\theta}_t}(u^{(i)})(y_j^{(i)})$ and $c_1, c_2 > 0$ are constants. Thus, the loss $\mathcal{L}(\boldsymbol{\theta})$ satisfies RSC w.r.t $(B_\kappa^t, \boldsymbol{\theta}_t)$ whenever $\alpha_t > 0$.

*Proof.* From the Taylor expansion in (15), to establish (19) it suffices to focus on the second order term and for $\boldsymbol{\theta}' \in B^t$ show

$$(\boldsymbol{\theta}' - \boldsymbol{\theta}_t)^\top \boldsymbol{H}(\tilde{\boldsymbol{\theta}})(\boldsymbol{\theta} - \boldsymbol{\theta}_t) \geq \alpha_t \|\boldsymbol{\theta}' - \boldsymbol{\theta}_t\|_2^2. \tag{29}$$

Given the $2 \times 2$ block structure of the Hessian as in (8), denoting $\delta\boldsymbol{\theta} := \boldsymbol{\theta}' - \boldsymbol{\theta}_t$ for compactness, we note that

$$\delta\boldsymbol{\theta}^\top \boldsymbol{H}(\tilde{\boldsymbol{\theta}})\delta\boldsymbol{\theta} = \underbrace{\delta\boldsymbol{\theta}_f^\top H_{ff}\delta\boldsymbol{\theta}_f}_{T_1} + \underbrace{2\delta\boldsymbol{\theta}_f^\top H_{fg}\delta\boldsymbol{\theta}_g}_{T_2} + \underbrace{\delta\boldsymbol{\theta}_g^\top H_{gg}\delta\boldsymbol{\theta}_g}_{T_3} . \tag{30}$$

Focusing on $T_1$ and using the exact form of $H_{ff}$ as in (11), we have

$$T_1 = \frac{1}{n}\sum_{i=1}^n \frac{1}{q_i}\sum_{j=1}^{q_i} \ell_{i,j}'' \left\langle \delta\boldsymbol{\theta}_f, \sum_{k=1}^K g_{k,j}^{(i)}\nabla_{\boldsymbol{\theta}_f}f_k^{(i)}\right\rangle^2 + \frac{1}{n}\sum_{i=1}^n \frac{1}{q_i}\sum_{j=1}^{q_i} \ell_{ij}' \sum_{k=1}^K g_{k,j}^{(i)}\delta\boldsymbol{\theta}_f^\top \nabla_{\boldsymbol{\theta}_f}^2 f_k^{(i)}\delta\boldsymbol{\theta}_f$$

$$\overset{(a)}{\geq} \frac{1}{n}\sum_{i=1}^n \frac{1}{q_i}\sum_{j=1}^{q_i} \ell_{i,j}'' \left\langle \delta\boldsymbol{\theta}_f, \nabla_{\boldsymbol{\theta}_f}G_{\boldsymbol{\theta}}(u^{(i)})(y_j^{(i)})\right\rangle^2 - \frac{\lambda c_0 c^{(f)}}{\sqrt{m_f}}\|\delta\boldsymbol{\theta}_f\|_2^2 , \tag{31}$$

where (a) follows from Assumption 2 and Lemma 4.1. The analysis for $T_3$ is similar, and we get

$$T_3 \geq \frac{1}{n}\sum_{i=1}^n \frac{1}{q_i}\sum_{j=1}^{q_i} \ell_{i,j}'' \left\langle \delta\boldsymbol{\theta}_g, \nabla_{\boldsymbol{\theta}_g}G_{\boldsymbol{\theta}}(u^{(i)})(y_j^{(i)})\right\rangle^2 - \frac{\lambda c_0 c^{(g)}}{\sqrt{m_g}}\|\delta\boldsymbol{\theta}_g\|_2^2 . \tag{32}$$

Focusing on $T_2$ and using the exact forms in terms of $H_{fg}^{(1)}$ and $H_{fg}^{(2)}$ as in (11), we have

$$\frac{1}{2}T_2 = \delta\boldsymbol{\theta}_f^\top \left(\frac{1}{n}\sum_{i=1}^n \frac{1}{q_i}\sum_{j=1}^{q_i} \ell_{ij}' \sum_{k=1}^K \nabla_{\boldsymbol{\theta}_f}f_k^{(i)}\nabla_{\boldsymbol{\theta}_g}g_{k,j}^{(i)\top}\right)\delta\boldsymbol{\theta}_g$$

$$+ \delta\boldsymbol{\theta}_f^\top \left(\frac{1}{n}\sum_{i=1}^n \frac{1}{q_i}\sum_{j=1}^{q_i} \ell_{i,j}''\left(\sum_{k=1}^K g_{k,j}^{(i)}\nabla_{\boldsymbol{\theta}_f}f_k^{(i)}\right)\left(\sum_{k'=1}^K f_{k'}^{(i)}\nabla_{\boldsymbol{\theta}_g}g_{k',j}^{(i)\top}\right)\right)\delta\boldsymbol{\theta}_g$$

$$\overset{(a)}{=} \delta\boldsymbol{\theta}_f^\top \left(\sum_{h=1}^{\tilde{q}}\sigma_h \boldsymbol{a}_h \boldsymbol{b}_h^\top\right)\delta\boldsymbol{\theta}_g + \delta\boldsymbol{\theta}_f^\top \left(\frac{1}{n}\sum_{i=1}^n \frac{1}{q_i}\sum_{j=1}^{q_i} \ell_{i,j}''\nabla_{\boldsymbol{\theta}_f}G_{\boldsymbol{\theta}}(u^{(i)})(y_j^{(i)})\nabla_{\boldsymbol{\theta}_g}G_{\boldsymbol{\theta}}(u^{(i)})(y_j^{(i)})^\top\right)\delta\boldsymbol{\theta}_g$$

$$= \sum_{h=1}^{\bar{q}}\sigma_h\langle\delta\boldsymbol{\theta}_f, \boldsymbol{a}_h\rangle\langle\delta\boldsymbol{\theta}_g, \boldsymbol{b}_h\rangle + \frac{1}{n}\sum_{i=1}^n \frac{1}{q_i}\sum_{j=1}^{q_i} \ell_{i,j}'' \left\langle \delta\boldsymbol{\theta}_f, \nabla_{\boldsymbol{\theta}_f}G_{\boldsymbol{\theta}}(u^{(i)})(y_j^{(i)})\right\rangle \left\langle \delta\boldsymbol{\theta}_g, \nabla_{\boldsymbol{\theta}_g}G_{\boldsymbol{\theta}}(u^{(i)})(y_j^{(i)})\right\rangle$$

$$\overset{(b)}{\geq} \frac{1}{n}\sum_{i=1}^n \frac{1}{q_i}\sum_{j=1}^{q_i} \ell_{i,j}'' \left\langle \delta\boldsymbol{\theta}_f, \nabla_{\boldsymbol{\theta}_f}G_{\boldsymbol{\theta}}(u^{(i)})(y_j^{(i)})\right\rangle \left\langle \delta\boldsymbol{\theta}_g, \nabla_{\boldsymbol{\theta}_g}G_{\boldsymbol{\theta}}(u^{(i)})(y_j^{(i)})\right\rangle . \tag{33}$$

where (a) follows from SVD as in (16), (b) follows since by Definition 3, $\langle\delta\boldsymbol{\theta}_f, \boldsymbol{a}_h\rangle \geq 0, \langle\delta\boldsymbol{\theta}_g, \boldsymbol{b}_h\rangle \geq 0$. Combining (31), (33), (32), using $m = m_g = m_f$ and $c_1 = \max(c^{(f)}, c^{(g)})$, we have

$$\delta\boldsymbol{\theta}^\top \boldsymbol{H}(\tilde{\boldsymbol{\theta}})\delta\boldsymbol{\theta} \geq \frac{1}{n}\sum_{i=1}^n \frac{1}{q_i}\sum_{j=1}^{q_i} \ell_{i,j}'' \left(\left\langle \delta\boldsymbol{\theta}_f, \nabla_{\boldsymbol{\theta}_f}G_{\boldsymbol{\theta}}(u^{(i)})(y_j^{(i)})\right\rangle + \left\langle \delta\boldsymbol{\theta}_g, \nabla_{\boldsymbol{\theta}_g}G_{\boldsymbol{\theta}}(u^{(i)})(y_j^{(i)})\right\rangle\right)^2 - \frac{\lambda c_0 c_1}{\sqrt{m}}\|\delta\boldsymbol{\theta}\|_2^2$$

$$\overset{(a)}{\geq} a\frac{1}{n}\sum_{i=1}^n \frac{1}{q_i}\sum_{j=1}^{q_i} \left(\left\langle \delta\boldsymbol{\theta}_f, \nabla_{\boldsymbol{\theta}_f}G_{\boldsymbol{\theta}}(u^{(i)})(y_j^{(i)})\right\rangle + \left\langle \delta\boldsymbol{\theta}_g, \nabla_{\boldsymbol{\theta}_g}G_{\boldsymbol{\theta}}(u^{(i)})(y_j^{(i)})\right\rangle\right)^2 - \frac{\lambda c_0 c_1}{\sqrt{m}}\|\delta\boldsymbol{\theta}\|_2^2$$

$$\overset{(b)}{\geq} a\left(\left\langle \delta\boldsymbol{\theta}_f, \nabla_{\boldsymbol{\theta}_f}\bar{G}_{\boldsymbol{\theta}}(u^{(i)})(y_j^{(i)})\right\rangle + \left\langle \delta\boldsymbol{\theta}_g, \nabla_{\boldsymbol{\theta}_g}\bar{G}_{\boldsymbol{\theta}}(u^{(i)})(y_j^{(i)})\right\rangle\right)^2 - \frac{\lambda c_0 c_1}{\sqrt{m}}\|\delta\boldsymbol{\theta}\|_2^2$$

$$= a\left\langle \delta\boldsymbol{\theta}, \nabla_{\boldsymbol{\theta}}\bar{G}_{\boldsymbol{\theta}}\right\rangle^2 - \frac{\lambda c_0 c_1}{\sqrt{m}}\|\delta\boldsymbol{\theta}\|_2^2$$

$$\overset{(c)}{\geq} a\kappa^2\|\nabla_{\boldsymbol{\theta}}\bar{G}_{\boldsymbol{\theta}}\|_2^2\|\delta\boldsymbol{\theta}\|_2^2 - \frac{\lambda c_0 c_1}{\sqrt{m}}\|\delta\boldsymbol{\theta}\|_2^2$$

$$= \alpha_t\|\delta\boldsymbol{\theta}\|_2^2 ,$$

where (a) follows from Assumption 2, (b) follows from Jensen's inequality and with $\bar{G}_{\boldsymbol{\theta}} = \frac{1}{n}\sum_{i=1}^n \frac{1}{q_i}\sum_{j=1}^{q_i} G_{\boldsymbol{\theta}}(u^{(i)})(y_j^{(i)})$ as in Definition 3, (c) follows from Definition 3, and $\alpha_t = a\kappa^2\|\nabla_{\boldsymbol{\theta}}\bar{G}_{\boldsymbol{\theta}}\|_2^2 - \frac{\lambda c_0 c_1}{\sqrt{m}}$. That completes the proof. $\qquad\square$

**Proposition 1** ($Q_\kappa^t$ is non-empty). *For over-parameterized branch and trunk nets with $p_f, p_g > qk$, the restricted set $Q_\kappa^t$ is non-empty.*

*Proof.* We simply construct a $\boldsymbol{\theta}' = [{\boldsymbol{\theta}'_f}^\top \; {\boldsymbol{\theta}'_g}^\top]^\top \in Q_\kappa^t$ along with the value of $\kappa$. Without loss of generality, we make $\boldsymbol{\theta}_t$ the origin of the coordinate system and work with the unit vector $\bar{\mathbf{g}} = [\bar{\mathbf{g}}_f^\top \; \bar{\mathbf{g}}_g^\top]^\top = \frac{\nabla_{\boldsymbol{\theta}} \bar{G}_{\boldsymbol{\theta}_t}}{\|\nabla_{\boldsymbol{\theta}} \bar{G}_{\boldsymbol{\theta}_t}\|_2}$. Further, we assume $\boldsymbol{\theta}'$ also to be a unit vector. Then, our problem reduces to feasibility of the following system of two quadratic equations over $\boldsymbol{\theta}'_f \in \mathbb{R}^{p_f}, \boldsymbol{\theta}'_g \in \mathbb{R}^{p_g}$:

$$\boldsymbol{\theta}'^\top_f \left( \sum_{h=1}^{\tilde{q}} \sigma_{h_1} \boldsymbol{a}_{h_1} \boldsymbol{b}_{h_1}^\top \right) \boldsymbol{\theta}'_g \geq 0$$

$$\left( \langle \boldsymbol{\theta}'_f, \bar{\mathbf{g}}_f \rangle + \langle \boldsymbol{\theta}'_g, \bar{\mathbf{g}}_g \rangle \right)^2 \geq \kappa^2 \, ,$$

where $\sigma_{h_1}, \sigma_{h_2} > 0$, $\boldsymbol{a}_h \in \mathbb{R}^{p_f}$ are orthogonal unit vectors, $\boldsymbol{b}_h \in \mathbb{R}^{p_g}$ are orthogonal unit vectors, $\bar{\mathbf{g}} = [\bar{\mathbf{g}}_f^\top \; \bar{\mathbf{g}}_g^\top]^\top \in \mathbb{R}^{p_f + p_g}$ and $\boldsymbol{\theta}' = [\boldsymbol{\theta}'_f; \boldsymbol{\theta}'_g] \in \mathbb{R}^{p_f + p_g}$ are unit vectors, and we can choose $\kappa \in (0, 1]$. Without loss of generality, assume $\|\bar{\mathbf{g}}_f\|_2 \geq \|\bar{\mathbf{g}}_g\|_2$. Then, set $\boldsymbol{\theta}'_g = \mathbf{0}$ so that our feasibility condition reduces to $\langle \boldsymbol{\theta}'_f, \bar{\mathbf{g}}_f \rangle^2 \geq \kappa^2$ for some suitably chosen $\kappa \in (0, 1]$. Finally, set $\boldsymbol{\theta}'_f = \frac{\bar{\mathbf{g}}_f}{\|\bar{\mathbf{g}}_f\|_2}$ so that

$$\langle \boldsymbol{\theta}'_f, \bar{\mathbf{g}}_f \rangle^2 = \left( \frac{\bar{\mathbf{g}}_f}{\|\bar{\mathbf{g}}_f\|_2} \bar{\mathbf{g}}_f \right)^2 = \|\bar{\mathbf{g}}_f\|_2^2 := \kappa^2 \, ,$$

so that $\kappa \in (0, 1]$ (in fact, $\kappa \geq 1/\sqrt{2}$) as desired. That completes the proof. $\qquad \square$

**Theorem 4.3** (Smoothness of Loss). Under the assumptions Assumptions 1, 2 and 3, with high probability, for $\boldsymbol{\theta} \in B_\rho^{\mathrm{Euc}}(\boldsymbol{\theta}_0)$, $\mathcal{L}(\boldsymbol{\theta})$ is $\beta$-smooth with $\beta = b\varrho^2 + \frac{c\sqrt{\lambda}}{\sqrt{m}}$ with $c = \max(c^{(f)}, c^{(g)})$, $\varrho = \max(\varrho^{(f)}, \varrho^{(g)})$ with $c^{(f)}, c^{(g)}, \varrho^{(f)}, \varrho^{(g)}$ as in Lemma 4.1.

*Proof.* By the second order Taylor expansion about $\bar{\boldsymbol{\theta}}$, we have $\mathcal{L}(\boldsymbol{\theta}') = \mathcal{L}(\bar{\boldsymbol{\theta}}) + \langle \boldsymbol{\theta}' - \bar{\boldsymbol{\theta}}, \nabla_{\boldsymbol{\theta}} \mathcal{L}(\bar{\boldsymbol{\theta}}) \rangle + \frac{1}{2}(\boldsymbol{\theta}' - \bar{\boldsymbol{\theta}})^\top \frac{\partial^2 \mathcal{L}(\tilde{\boldsymbol{\theta}})}{\partial \boldsymbol{\theta}^2}(\boldsymbol{\theta}' - \bar{\boldsymbol{\theta}})$, where $\tilde{\boldsymbol{\theta}} = \xi \boldsymbol{\theta}' + (1 - \xi)\bar{\boldsymbol{\theta}}$ for some $\xi \in [0, 1]$. Then,

$$(\boldsymbol{\theta}' - \bar{\boldsymbol{\theta}})^\top \frac{\partial^2 \mathcal{L}(\tilde{\boldsymbol{\theta}})}{\partial \boldsymbol{\theta}^2}(\boldsymbol{\theta}' - \bar{\boldsymbol{\theta}}) = (\boldsymbol{\theta}' - \bar{\boldsymbol{\theta}})^\top \left( \frac{1}{n} \sum_{i=1}^n \frac{1}{q_i} \sum_{j=1}^{q_i} \ell''_{i,j} \nabla G_{\tilde{\boldsymbol{\theta}}}(u^{(i)})(y_j^{(i)}) \nabla G_{\tilde{\boldsymbol{\theta}}}(u^{(i)})(y_j^{(i)})^\top \right.$$

$$\left. + \ell'_{i,j} \nabla^2 G_{\tilde{\boldsymbol{\theta}}}(u^{(i)})(y_j^{(i)}) \right)(\boldsymbol{\theta}' - \bar{\boldsymbol{\theta}})$$

$$= \underbrace{\frac{1}{n} \sum_{i=1}^n \frac{1}{q_i} \sum_{j=1}^{q_i} \ell''_{i,j} \left\langle \boldsymbol{\theta}' - \bar{\boldsymbol{\theta}}, \nabla G_{\tilde{\boldsymbol{\theta}}}(u^{(i)})(y_j^{(i)}) \right\rangle^2}_{I_1}$$

$$+ \underbrace{\frac{1}{n} \sum_{i=1}^n \frac{1}{q_i} \sum_{j=1}^{q_i} \ell'_{i,j} (\boldsymbol{\theta}' - \bar{\boldsymbol{\theta}})^\top \nabla^2 G_{\tilde{\boldsymbol{\theta}}}(u^{(i)})(y_j^{(i)})(\boldsymbol{\theta}' - \bar{\boldsymbol{\theta}})}_{I_2} \, .$$

Now, note that

$$I_1 = \frac{1}{n} \sum_{i=1}^n \frac{1}{q_i} \sum_{j=1}^{q_i} \ell''_{i,j} \left\langle \boldsymbol{\theta}' - \bar{\boldsymbol{\theta}}, \nabla G_{\tilde{\boldsymbol{\theta}}}(u^{(i)})(y_j^{(i)}) \right\rangle^2$$

$$\overset{(a)}{\leq} \frac{b}{n} \sum_{i=1}^n \frac{1}{q_i} \sum_{j=1}^{q_i} \left\| \nabla G_{\tilde{\boldsymbol{\theta}}}(u^{(i)})(y_j^{(i)}) \right\|_2^2 \|\boldsymbol{\theta}' - \bar{\boldsymbol{\theta}}\|_2^2$$

$$\overset{(b)}{\leq} b\varrho^2 \|\boldsymbol{\theta}' - \bar{\boldsymbol{\theta}}\|_2^2 \, ,$$

where (a) follows by the Cauchy-Schwartz inequality and (b) from Lemma 4.1.

For $I_2$, with $Q_{t,(i,j)} = (\boldsymbol{\theta}' - \bar{\boldsymbol{\theta}})^\top \nabla^2 G_{\tilde{\boldsymbol{\theta}}}(u^{(i)})(y_j^{(i)})(\boldsymbol{\theta}' - \bar{\boldsymbol{\theta}})$, we have

$$|Q_{t,(i,j)}| \leq \|\boldsymbol{\theta}' - \bar{\boldsymbol{\theta}}\|_2^2 \left\| \nabla^2 G_{\tilde{\boldsymbol{\theta}}}(u^{(i)})(y_j^{(i)}) \right\|_2 \leq \frac{c\|\boldsymbol{\theta}' - \bar{\boldsymbol{\theta}}\|_2^2}{\sqrt{m}} \ .$$

Then, we have

$$
\begin{aligned}
I_2 &= \frac{1}{n} \sum_{i=1}^n \frac{1}{q_i} \sum_{j=1}^{q_i} \ell'_{i,j} (\boldsymbol{\theta}' - \bar{\boldsymbol{\theta}})^\top \nabla^2 G_{\tilde{\boldsymbol{\theta}}}(u^{(i)})(y_j^{(i)})(\boldsymbol{\theta}' - \bar{\boldsymbol{\theta}}) \\
&\overset{(a)}{\leq} \lambda \left( \frac{1}{n} \sum_{i=1}^n \frac{1}{q_i} Q_{t,(i,j)}^2 \right)^{1/2} \\
&\leq \lambda \frac{c\|\boldsymbol{\theta}' - \bar{\boldsymbol{\theta}}\|_2^2}{\sqrt{m}} \ ,
\end{aligned}
$$

where (a) follows by Cauchy-Schwartz. Putting the upper bounds on $I_1$ and $I_2$ back, we have

$$(\boldsymbol{\theta}' - \bar{\boldsymbol{\theta}})^\top \nabla^2 G_{\tilde{\boldsymbol{\theta}}}(u^{(i)})(y_j^{(i)})(\boldsymbol{\theta}' - \bar{\boldsymbol{\theta}}) \leq \left[ b\varrho^2 + \frac{c\sqrt{\lambda_t}}{\sqrt{m}} \right] \|\boldsymbol{\theta}' - \bar{\boldsymbol{\theta}}\|_2^2 \ .$$

This completes the proof. $\qquad\square$

**Lemma 4.4** (RSC $\implies$ Restricted PL). The RSC and Smoothness of the Loss together imply a form of Polyak-Łojasiewicz (PL) condition w.r.t. the tuple $(B_t, \boldsymbol{\theta}_t)$, unlike standard PL which holds without restrictions (Karimi et al., 2016).

*Proof.* Define

$$\hat{\mathcal{L}}_{\boldsymbol{\theta}_t}(\boldsymbol{\theta}) := \mathcal{L}(\boldsymbol{\theta}_t) + \langle \boldsymbol{\theta} - \boldsymbol{\theta}_t, \nabla_{\boldsymbol{\theta}} \mathcal{L}(\boldsymbol{\theta}_t) \rangle + \frac{\alpha_t}{2} \|\boldsymbol{\theta} - \boldsymbol{\theta}_t\|_2^2 \ .$$

By Theorem A.3, $\forall \boldsymbol{\theta}' \in B_t$, we have

$$\mathcal{L}(\boldsymbol{\theta}') \geq \hat{\mathcal{L}}_{\boldsymbol{\theta}_t}(\boldsymbol{\theta}') \ . \tag{34}$$

Further, note that $\hat{\mathcal{L}}_{\boldsymbol{\theta}_t}(\boldsymbol{\theta})$ is minimized at $\hat{\boldsymbol{\theta}}_{t+1} := \boldsymbol{\theta}_t - \nabla_{\boldsymbol{\theta}} \mathcal{L}(\boldsymbol{\theta}_t)/\alpha_t$ and the minimum value is:

$$\inf_{\boldsymbol{\theta}} \hat{\mathcal{L}}_{\boldsymbol{\theta}_t}(\boldsymbol{\theta}) = \hat{\mathcal{L}}_{\boldsymbol{\theta}_t}(\hat{\boldsymbol{\theta}}_{t+1}) = \mathcal{L}(\boldsymbol{\theta}_t) - \frac{1}{2\alpha_t} \|\nabla_{\boldsymbol{\theta}} \mathcal{L}(\boldsymbol{\theta}_t)\|_2^2 \ .$$

Then, we have

$$\inf_{\boldsymbol{\theta} \in B_t} \mathcal{L}(\boldsymbol{\theta}) \overset{(a)}{\geq} \inf_{\boldsymbol{\theta} \in B_t} \hat{\mathcal{L}}_{\boldsymbol{\theta}_t}(\boldsymbol{\theta}) \geq \inf_{\boldsymbol{\theta}} \hat{\mathcal{L}}_{\boldsymbol{\theta}_t}(\boldsymbol{\theta}) = \mathcal{L}(\boldsymbol{\theta}_t) - \frac{1}{2\alpha_t} \|\nabla_{\boldsymbol{\theta}} \mathcal{L}(\boldsymbol{\theta}_t)\|_2^2 \ ,$$

where (a) follows from (34). Rearranging terms completes the proof. $\qquad\square$

**Lemma 1** (Local Loss Reduction in $B_t$). *Let $\alpha_t, \beta$ be as in Theorems A.3 and A.3 respectively, and $B_t := Q_\kappa^t \cap B_\rho^{\mathrm{Euc}}(\boldsymbol{\theta}_0) \cap B_{\rho_2}^{\mathrm{Euc}}(\boldsymbol{\theta}_t)$. Under assumptions Assumptions 1, 2 and 3, for gradient descent with step size $\eta_t = \frac{\omega_t}{\beta}, \omega_t \in (0, 2)$, for any $\overline{\boldsymbol{\theta}}_{t+1} \in \arg\inf_{\boldsymbol{\theta} \in B_t} \mathcal{L}(\boldsymbol{\theta})$, we have with high probability*

$$\mathcal{L}(\boldsymbol{\theta}_{t+1}) - \mathcal{L}(\overline{\boldsymbol{\theta}}_{t+1}) \leq \left( 1 - \frac{\alpha_t \omega_t}{\beta}(2 - \omega_t) \right) (\mathcal{L}(\boldsymbol{\theta}_t) - \mathcal{L}(\overline{\boldsymbol{\theta}}_{t+1})) \ . \tag{35}$$

*Proof.* Since $\mathcal{L}$ is $\beta$-smooth by Theorem A.3, we have

$$
\begin{aligned}
\mathcal{L}(\boldsymbol{\theta}_{t+1}) &\leq \mathcal{L}(\boldsymbol{\theta}_t) + \langle \boldsymbol{\theta}_{t+1} - \boldsymbol{\theta}_t, \nabla_{\boldsymbol{\theta}} \mathcal{L}(\boldsymbol{\theta}_t) \rangle + \frac{\beta}{2} \|\boldsymbol{\theta}_{t+1} - \boldsymbol{\theta}_t\|_2^2 \\
&= \mathcal{L}(\boldsymbol{\theta}_t) - \eta_t \|\nabla_{\boldsymbol{\theta}} \mathcal{L}(\boldsymbol{\theta}_t)\|_2^2 + \frac{\beta\eta_t^2}{2} \|\nabla_{\boldsymbol{\theta}} \mathcal{L}(\boldsymbol{\theta}_t)\|_2^2 \\
&= \mathcal{L}(\boldsymbol{\theta}_t) - \eta_t \left( 1 - \frac{\beta\eta_t}{2} \right) \|\nabla_{\boldsymbol{\theta}} \mathcal{L}(\boldsymbol{\theta}_t)\|_2^2
\end{aligned} \tag{36}
$$

Since $\bar{\boldsymbol{\theta}}_{t+1} \in \operatorname{arginf}_{\boldsymbol{\theta} \in B_t} \mathcal{L}(\boldsymbol{\theta})$ and $\alpha_t > 0$ by assumption, from Lemma A.3 we obtain

$$-\|\nabla_{\boldsymbol{\theta}} \mathcal{L}(\boldsymbol{\theta}_t)\|_2^2 \leq -2\alpha_t(\mathcal{L}(\boldsymbol{\theta}_t) - \mathcal{L}(\bar{\boldsymbol{\theta}}_{t+1})) .$$

Hence

$$
\begin{aligned}
\mathcal{L}(\boldsymbol{\theta}_{t+1}) - \mathcal{L}(\bar{\boldsymbol{\theta}}_{t+1}) &\leq \mathcal{L}(\boldsymbol{\theta}_t) - \mathcal{L}(\bar{\boldsymbol{\theta}}_{t+1}) - \eta_t \left(1 - \frac{\beta\eta_t}{2}\right) \|\nabla_{\boldsymbol{\theta}} \mathcal{L}(\boldsymbol{\theta}_t)\|_2^2 \\
&\overset{(a)}{\leq} \mathcal{L}(\boldsymbol{\theta}_t) - \mathcal{L}(\bar{\boldsymbol{\theta}}_{t+1}) - \eta_t \left(1 - \frac{\beta\eta_t}{2}\right) 2\alpha_t(\mathcal{L}(\boldsymbol{\theta}_t) - \mathcal{L}(\bar{\boldsymbol{\theta}}_{t+1})) \\
&= \left(1 - 2\alpha_t \eta_t \left(1 - \frac{\beta\eta_t}{2}\right)\right)(\mathcal{L}(\boldsymbol{\theta}_t) - \mathcal{L}(\bar{\boldsymbol{\theta}}_{t+1}))
\end{aligned}
$$

where (a) follows for any $\eta_t \leq \frac{2}{\beta}$ because this implies $1 - \frac{\beta\eta_t}{2} \geq 0$. Choosing $\eta_t = \frac{\omega_T}{\beta}, \omega_t \in (0, 2)$,

$$\mathcal{L}(\boldsymbol{\theta}_{t+1}) - \mathcal{L}(\bar{\boldsymbol{\theta}}_{t+1}) \leq \left(1 - \frac{\alpha_t \omega_t}{\beta}(2 - \omega_t)\right)(\mathcal{L}(\boldsymbol{\theta}_t) - \mathcal{L}(\bar{\boldsymbol{\theta}}_{t+1})) .$$

This completes the proof. $\qquad\square$

**Theorem 4.5** (Global Loss Reduction). Consider the same conditions as in Theorem 4.3 and $\alpha_t > 0$ for $t \in [T]$ for a gradient descent update $\boldsymbol{\theta}_{t+1} = \boldsymbol{\theta}_t - \eta_t \nabla_{\boldsymbol{\theta}} \mathcal{L}(\boldsymbol{\theta}_t)$ with $\eta_t = \frac{\omega_t}{\beta}$ for some $\omega_t \in (0, 2)$, where $\beta$ is defined as in Theorem 4.3. Then, with high probability, $\forall \bar{\boldsymbol{\theta}} \in \operatorname*{arginf}_{\boldsymbol{\theta} \in B_\rho^{\mathrm{Euc}}(\boldsymbol{\theta}_0)} \mathcal{L}(\boldsymbol{\theta})$ with

$0 \leq \gamma_t := \frac{\mathcal{L}(\bar{\boldsymbol{\theta}}_{t+1}) - \mathcal{L}(\bar{\boldsymbol{\theta}})}{\mathcal{L}(\boldsymbol{\theta}_t) - \mathcal{L}(\bar{\boldsymbol{\theta}})} < 1$ and $\bar{\boldsymbol{\theta}}_{t+1} \in \operatorname{arginf}_{\boldsymbol{\theta} \in Q_\kappa^t \cap B_\rho^{\mathrm{Euc}}(\boldsymbol{\theta}_0)} \mathcal{L}(\boldsymbol{\theta})$, we have

$$\mathcal{L}(\boldsymbol{\theta}_{t+1}) - \mathcal{L}(\bar{\boldsymbol{\theta}}) \leq \left(1 - \frac{\alpha_t \omega_t (1 - \gamma_t)}{\beta}(2 - \omega_t)\right)(\mathcal{L}(\boldsymbol{\theta}_t) - \mathcal{L}(\bar{\boldsymbol{\theta}})) . \tag{20}$$

*Proof.* We start by showing $\gamma_t = \frac{\mathcal{L}(\bar{\boldsymbol{\theta}}_{t+1}) - \mathcal{L}(\boldsymbol{\theta}^*)}{\mathcal{L}(\boldsymbol{\theta}_t) - \mathcal{L}(\boldsymbol{\theta}^*)}$ satisfies $0 \leq \gamma_t < 1$. Since $\boldsymbol{\theta}^* \in \operatorname*{arginf}_{\boldsymbol{\theta} \in B_\rho^{\mathrm{Euc}}(\boldsymbol{\theta}_0)} \mathcal{L}(\boldsymbol{\theta})$,

$\bar{\boldsymbol{\theta}}_{t+1} \in \operatorname*{arginf}_{\boldsymbol{\theta} \in B_t} \mathcal{L}(\boldsymbol{\theta})$, and $\boldsymbol{\theta}_{t+1} \in Q_\kappa^t \cap B_\rho^{\mathrm{Euc}}(\boldsymbol{\theta}_0)$ by the definition of gradient descent, we have

$$\mathcal{L}(\boldsymbol{\theta}^*) \leq \mathcal{L}(\bar{\boldsymbol{\theta}}_{t+1}) \leq \mathcal{L}(\boldsymbol{\theta}_{t+1}) \overset{(a)}{\leq} \mathcal{L}(\boldsymbol{\theta}_t) - \frac{1}{2\beta}\|\nabla_{\boldsymbol{\theta}} \mathcal{L}(\boldsymbol{\theta}_t)\|_2^2 < \mathcal{L}(\boldsymbol{\theta}_t) ,$$

where (a) follows from (36). Since $\mathcal{L}(\bar{\boldsymbol{\theta}}_{t+1}) \geq \mathcal{L}(\boldsymbol{\theta}^*)$ and $\mathcal{L}(\boldsymbol{\theta}_t) > \mathcal{L}(\boldsymbol{\theta}^*)$, we have $\gamma_t \geq 0$. Further, since $\mathcal{L}(\bar{\boldsymbol{\theta}}_{t+1}) < \mathcal{L}(\boldsymbol{\theta}_t)$, we have $\gamma_t < 1$.

Now, with $\omega_t \in (0, 2)$, we have

$$
\begin{aligned}
\mathcal{L}(\boldsymbol{\theta}_{t+1}) - \mathcal{L}(\boldsymbol{\theta}^*) &= \mathcal{L}(\boldsymbol{\theta}_{t+1}) - \mathcal{L}(\bar{\boldsymbol{\theta}}_{t+1}) + \mathcal{L}(\bar{\boldsymbol{\theta}}_{t+1}) - \mathcal{L}(\boldsymbol{\theta}^*) \\
&\leq \left(1 - \frac{\alpha_t \omega_t}{\beta}(2 - \omega_t)\right)(\mathcal{L}(\boldsymbol{\theta}_t) - \mathcal{L}(\bar{\boldsymbol{\theta}}_{t+1})) + \left(1 - \frac{\alpha_t \omega_t}{\beta}(2 - \omega_t)\right)(\mathcal{L}(\bar{\boldsymbol{\theta}}_{t+1}) - \mathcal{L}(\boldsymbol{\theta}^*)) \\
&\quad + \left(\mathcal{L}(\bar{\boldsymbol{\theta}}_{t+1}) - \left(1 - \frac{\alpha_t \omega_t}{\beta}(2 - \omega_t)\right)\mathcal{L}(\bar{\boldsymbol{\theta}}_{t+1})\right) - \left(\mathcal{L}(\boldsymbol{\theta}^*) - \left(1 - \frac{\alpha_t \omega_t}{\beta}(2 - \omega_t)\right)\mathcal{L}(\boldsymbol{\theta}^*)\right) \\
&= \left(1 - \frac{\alpha_t \omega_t}{\beta}(2 - \omega_t)\right)(\mathcal{L}(\boldsymbol{\theta}_t) - \mathcal{L}(\boldsymbol{\theta}^*)) + \frac{\alpha_t \omega_t}{\beta}(2 - \omega_t)(\mathcal{L}(\bar{\boldsymbol{\theta}}_{t+1}) - \mathcal{L}(\boldsymbol{\theta}^*)) \\
&= \left(1 - \frac{\alpha_t \omega_t}{\beta}(2 - \omega_t)\right)(\mathcal{L}(\boldsymbol{\theta}_t) - \mathcal{L}(\boldsymbol{\theta}^*)) + \frac{\alpha_t \omega_t}{\beta}(2 - \omega_t)\gamma_t(\mathcal{L}(\boldsymbol{\theta}_t) - \mathcal{L}(\boldsymbol{\theta}^*)) \\
&= \left(1 - \frac{\alpha_t \omega_t}{\beta}(1 - \gamma_t)(2 - \omega_t)\right)(\mathcal{L}(\boldsymbol{\theta}_t) - \mathcal{L}(\boldsymbol{\theta}^*)) .
\end{aligned}
$$

That completes the proof. $\qquad\square$

### A.4 OPTIMIZATION GUARANTEES FOR DEEPONETS: RELU ACTIVATIONS

Recall the DeepONet predictor (1)

$$G_{\boldsymbol{\theta}}\left(u^{(i)}\right)(y_j^{(i)}) := \sum_{k=1}^{K} f_k\left(\boldsymbol{\theta}_f; u^{(i)}\right) g_k\left(\boldsymbol{\theta}_g; y_j^{(i)}\right) . \tag{37}$$

In the analysis, it is useful to distinguish between the parameters up to the pre-final layer and the final layer, i.e. $\dim(\boldsymbol{\theta}) = M_f + M_g + (m_f + m_g)K$, where $M_f$ and $M_g$ denote the number of parameters in the branch and trunk nets till the pre-final layer respectively and $m_f \cdot K$ and $m_g \cdot K$ are the number of weights in the last layer of the branch and trunk nets respectively. In essence we have $\dim(\boldsymbol{\theta}_f) = M_f + m_f \cdot K$ and $\dim(\boldsymbol{\theta}_g) = M_g + m_g \cdot K$. We note that it is sufficient to show positive definiteness of the above NTK at initialization. Once that has been established, standard approaches (Jacot et al., 2018; Du et al., 2019; Arora et al., 2019b;a; Allen-Zhu et al., 2019) allow one to show the geometric convergence of (S)GD. In the sequel it proves useful to rewrite the branch and trunk net outputs as:

$$f_k\left(\boldsymbol{\theta}_f; u^{(i)}\right) = \sum_{h=1}^{m_f} w_{k,h}^{(f)} \bar{f}_h\left(\boldsymbol{\theta}_{\bar{f}}; u^{(i)}\right) , \quad g_k\left(\boldsymbol{\theta}_g; y_j^{(i)}\right) = \sum_{h'=1}^{m_g} w_{k,h'}^{(g)} \bar{g}_{h'}\left(\boldsymbol{\theta}_{\bar{g}}; y_j^{(i)}\right) , \quad \forall\, k \in [K],$$
$$\tag{38}$$

where $w_{k,h}^{(f)}, h \in [m_f]$ are the weights of the linear last layer of the branch net and $w_{k,h'}^{(g)}, h' \in [m_g]$ are the weights of the linear last layer of the trunk net. Similarly, $\boldsymbol{\theta}_{\bar{f}}$ and $\boldsymbol{\theta}_{\bar{g}}$ are the parameters leading up to the pre-final layer in branch net with $m_f$ outputs $[\bar{f}_h, h \in [m_f]]$ and trunk net with $m_g$ outputs $[\bar{g}_{h'}, h' \in [m_g]]$ respectively. We will denote by $\boldsymbol{\theta}_{f,k}$ all the parameters corresponding to $f_k$, i.e. $\boldsymbol{\theta}_{f,k} := \{w_{k,h}^{(f)}, h \in [m_f], \boldsymbol{\theta}_{\bar{f}}\}$ which includes all the parameters needed for $f_k$ for each $k \in [K]$. Similarly we denote by $\boldsymbol{\theta}_{g,k}$, all the parameters corresponding to $g_k$, i.e. $\boldsymbol{\theta}_{g,k} := \{w_{k,h'}^{(g)}, h' \in [m_g], \boldsymbol{\theta}_{\bar{g}}\}$.

We can explicitly write the NTK for the DeepONet model, specifically entry corresponding to the inputs $\{u^{(i)}, y_j^{(i)}\}$ and $\{u^{(i')}, y_{j'}^{(i')}\}$ as:

$$\left\langle \nabla_{\boldsymbol{\theta}} G_{\boldsymbol{\theta}}(u^{(i)})(y_j^{(i)}) , \nabla_{\boldsymbol{\theta}} G_{\boldsymbol{\theta}}(u^{(i')})(y_{j'}^{(i')}) \right\rangle$$

$$= \sum_{k,k'=1}^{K} g_k\left(\boldsymbol{\theta}_g; y_j^{(i)}\right) g_{k'}\left(\boldsymbol{\theta}_g; y_{j'}^{(i')}\right) \left\langle \nabla_{\boldsymbol{\theta}_f} f_k\left(\boldsymbol{\theta}_f; u^{(i)}\right) , \nabla_{\boldsymbol{\theta}_f} f_{k'}\left(\boldsymbol{\theta}_f; u^{(i')}\right) \right\rangle$$

$$+ \sum_{k,k'=1}^{K} f_k\left(\boldsymbol{\theta}_f; u^{(i)}\right) f_{k'}\left(\boldsymbol{\theta}_f; u^{(i')}\right) \left\langle \nabla_{\boldsymbol{\theta}_g} g_k\left(\boldsymbol{\theta}_g; y_j^{(i)}\right) , \nabla_{\boldsymbol{\theta}_g} g_{k'}\left(\boldsymbol{\theta}_g; y_{j'}^{(i')}\right) \right\rangle$$

$$= \sum_{k=1}^{K} \underbrace{g_k\left(\boldsymbol{\theta}_g; y_j^{(i)}\right) g_k\left(\boldsymbol{\theta}_g; y_{j'}^{(i')}\right)}_{T_k^{(1)}} \left\langle \nabla_{\boldsymbol{\theta}_f} f_k\left(\boldsymbol{\theta}_f; u^{(i)}\right) , \nabla_{\boldsymbol{\theta}_f} f_k\left(\boldsymbol{\theta}_f; u^{(i')}\right) \right\rangle$$

$$+ \sum_{k=1}^{K} \underbrace{f_k\left(\boldsymbol{\theta}_f; u^{(i)}\right) f_k\left(\boldsymbol{\theta}_f; u^{(i')}\right)}_{T_k^{(2)}} \left\langle \nabla_{\boldsymbol{\theta}_g} g_k\left(\boldsymbol{\theta}_g; y_j^{(i)}\right) , \nabla_{\boldsymbol{\theta}_g} g_k\left(\boldsymbol{\theta}_g; y_{j'}^{(i')}\right) \right\rangle$$

$$+ \sum_{\substack{k,k'=1 \\ k \neq k'}}^{K} \underbrace{g_k\left(\boldsymbol{\theta}_g; y_j^{(i)}\right) g_{k'}\left(\boldsymbol{\theta}_g; y_{j'}^{(i')}\right)}_{T_{k,k'}^{(3)}} \left\langle \nabla_{\boldsymbol{\theta}_f} f_k\left(\boldsymbol{\theta}_f; u^{(i)}\right) , \nabla_{\boldsymbol{\theta}_f} f_{k'}\left(\boldsymbol{\theta}_f; u^{(i')}\right) \right\rangle$$

$$+ \sum_{\substack{k,k'=1 \\ k \neq k'}}^{K} \underbrace{f_k\left(\boldsymbol{\theta}_f; u^{(i)}\right) f_{k'}\left(\boldsymbol{\theta}_f; u^{(i')}\right)}_{T_{k,k'}^{(4)}} \left\langle \nabla_{\boldsymbol{\theta}_g} g_k\left(\boldsymbol{\theta}_g; y_j^{(i)}\right) , \nabla_{\boldsymbol{\theta}_g} g_{k'}\left(\boldsymbol{\theta}_g; y_{j'}^{(i')}\right) \right\rangle .$$

$$\tag{39}$$

**Theorem 5.1** ($\mathcal{K}(\boldsymbol{\theta})$ is positive definite at initialization). Given standard initialization for the branch and trunk nets, and granted that the individual branch and trunk net NTKs are positive definite (24)-(25), the NTK of DeepONet is positive definite at initialization, i.e.

$$\boldsymbol{\alpha}^{\top}\mathbb{E}[\mathcal{K}(\boldsymbol{\theta})]\boldsymbol{\alpha} \geq c > 0, \tag{26}$$

where $\boldsymbol{\alpha}$ denotes an arbitrary block unit vector with $n$ blocks, and $\alpha_{i,j}$ corresponds to the $j$-th entry in the $i$-th block and $c$ denotes a positive constant.

*Proof.* The proof follows as a direct consequence of Proposition 2 together with Propositions 3 and 4 □

**Proposition 2.** *With the branch and trunk net weights initialized using standard initialization techniques and the last layer of branch layer initialized according to Assumption 3, we have*

$$\mathbb{E}[T_{k,k'}^{(3)}|\boldsymbol{\theta}_{\bar{g}},\boldsymbol{\theta}_{\bar{g}}] = 0, \quad \mathbb{E}[T_{k,k'}^{(4)}|\boldsymbol{\theta}_{\bar{g}},\boldsymbol{\theta}_{\bar{g}}] = 0, \quad k,k' \in [K]. \tag{40}$$

*where $T_{k,k'}^{(3)}$ is defined in (39)*

*Proof.* As noted in Assumption 3, the last layer weights for the branch and trunk nets are initialized as zero mean Gaussians, i.e., $w_{k,h}^{(f)}, w_{k,h'}^{(g)} \sim \mathcal{N}(0, \frac{1}{mK})$ for $k \in [K], h \in [m_f], h' \in [m_g]$, similar to the other layers. Now,

$$\begin{aligned} T_{k,k'}^{(3)} &= g_k\left(\boldsymbol{\theta}_g; y_j^{(i)}\right) g_{k'}\left(\boldsymbol{\theta}_g; y_{j'}^{(i')}\right) \\ &= \left(\sum_{h'=1}^{m_g} w_{k,h'}^{(g)} \bar{g}_{h'}\left(\boldsymbol{\theta}_{\bar{g}}; y_j^{(i)}\right)\right)\left(\sum_{\tilde{h}'=1}^{m_g} w_{k',\tilde{h}'}^{(g)} \bar{g}_{\tilde{h}'}\left(\boldsymbol{\theta}_{\bar{g}}; y_{j'}^{(i')}\right)\right) \\ &= \sum_{h',\tilde{h}'=1}^{m_g} w_{k,h'}^{(g)} w_{k',\tilde{h}'}^{(g)} \bar{g}_{h'}\left(\boldsymbol{\theta}_{\bar{g}}; y_j^{(i)}\right) \bar{g}_{\tilde{h}'}\left(\boldsymbol{\theta}_{\bar{g}}; y_{j'}^{(i')}\right), \end{aligned} \tag{41}$$

which in turn implies

$$\begin{aligned} \mathbb{E}[T_{k,k'}^{(3)}|\boldsymbol{\theta}_{\bar{g}},\boldsymbol{\theta}_{\bar{g}}] &= \sum_{h',\tilde{h}'=1}^{m_g} \mathbb{E}[w_{k,h'}^{(g)} w_{k',\tilde{h}'}^{(g)}] \bar{g}_{h'}\left(\boldsymbol{\theta}_{\bar{g}}; y_j^{(i)}\right) \bar{g}_{\tilde{h}'}\left(\boldsymbol{\theta}_{\bar{g}}; y_{j'}^{(i')}\right) \\ &\stackrel{(a)}{=} \sum_{h',\tilde{h}'=1}^{m_g} \mathbb{E}[w_{k,h'}^{(g)}]\mathbb{E}[w_{k',\tilde{h}'}^{(g)}] \bar{g}_{h'}\left(\boldsymbol{\theta}_{\bar{g}}; y_j^{(i)}\right) \bar{g}_{\tilde{h}'}\left(\boldsymbol{\theta}_{\bar{g}}; y_{j'}^{(i')}\right) \\ &= 0, \end{aligned} \tag{42}$$

where (a) follows since $w_{k,h'}^{(g)}$ and $w_{k',\tilde{h}'}^{(g)}$ are independent. The analysis for $T_{k,k'}^{(4)}$ is similar. This completes the proof. □

**Proposition 3.** *Given that the terms $T_{k,k'}^{(3)}$ and $T_{k,k'}^{(4)}$ can be suitably bounded close to zero, we have for any arbitrary block unit vector $\boldsymbol{\alpha}$*

$$\boldsymbol{\alpha}^T \mathcal{K}(\boldsymbol{\theta})\boldsymbol{\alpha} \geq \lambda_{0,f} \sum_{k=1}^{K} \left\|\boldsymbol{\alpha} \odot g_k(\boldsymbol{\theta}_g; \boldsymbol{y}^{(\cdot)})\right\|_2^2 + \lambda_{0,g} \sum_{k=1}^{K} \left\|\boldsymbol{\alpha} \odot \left(\mathbf{1}_p \otimes f_k(\boldsymbol{\theta}_f; \boldsymbol{u}^{(\cdot)})\right)\right\|_2^2. \tag{43}$$

*where $\mathbf{1}_q$ denotes the $q$-dimensional vector of all entries equal to one.*

*Proof.* Focusing on a quadratic form of $\mathcal{K}(\boldsymbol{\theta})$, and ignoring the $T^{(3)}, T^{(4)}_{k,k'}$ terms, for any arbitrary block unit vector $\boldsymbol{\alpha}$, we have

$$
\begin{aligned}
\boldsymbol{\alpha}^T \mathcal{K}(\boldsymbol{\theta}) \boldsymbol{\alpha} &= \sum_{(i,j),(i',j')} \alpha_{i,j} \alpha_{i',j'} \left\langle \nabla_{\boldsymbol{\theta}} G_{\boldsymbol{\theta}}(u^{(i)})(y_j^{(i)}) \ , \ \nabla_{\boldsymbol{\theta}} G_{\boldsymbol{\theta}}(u^{(i')})(y_{j'}^{(i')}) \right\rangle \\
&= \sum_{(i,j),(i',j')} \alpha_{i,j} \alpha_{i',j'} \sum_{k=1}^{K} g_k\left(\boldsymbol{\theta}_g; y_j^{(i)}\right) g_k\left(\boldsymbol{\theta}_g; y_{j'}^{(i')}\right) \left\langle \nabla_{\boldsymbol{\theta}_f} f_k\left(\boldsymbol{\theta}_f; u^{(i)}\right) \ , \ \nabla_{\boldsymbol{\theta}_f} f_k\left(\boldsymbol{\theta}_f; u^{(i')}\right) \right\rangle \\
&\quad + \sum_{(i,j),(i',j')} \alpha_{i,j} \alpha_{i',j'} \sum_{k=1}^{K} f_k\left(\boldsymbol{\theta}_f; u^{(i)}\right) f_{k'}\left(\boldsymbol{\theta}_f; u^{(i')}\right) \left\langle \nabla_{\boldsymbol{\theta}_g} g_k\left(\boldsymbol{\theta}_g; y_j^{(i)}\right) \ , \ \nabla_{\boldsymbol{\theta}_g} g_k\left(\boldsymbol{\theta}_g; y_{j'}^{(i')}\right) \right\rangle \\
&= \sum_{k=1}^{K} \sum_{(i,j),(i',j')} \alpha_{i,j} \alpha_{i',j'} g_k\left(\boldsymbol{\theta}_g; y_j^{(i)}\right) g_k\left(\boldsymbol{\theta}_g; y_{j'}^{(i')}\right) \left\langle \nabla_{\boldsymbol{\theta}} f_k\left(\boldsymbol{\theta}_f; u^{(i)}\right) \ , \ \nabla_{\boldsymbol{\theta}} f_k\left(\boldsymbol{\theta}_f; u^{(i')}\right) \right\rangle \\
&\quad + \sum_{k=1}^{K} \sum_{(i,j),(i',j')} \alpha_{i,j} \alpha_{i',j'} f_k\left(\boldsymbol{\theta}_f; u^{(i)}\right) f_k\left(\boldsymbol{\theta}_f; u^{(i')}\right) \left\langle \nabla_{\boldsymbol{\theta}} g_k\left(\boldsymbol{\theta}_g; y_j^{(i)}\right) \ , \ \nabla_{\boldsymbol{\theta}} g_k\left(\boldsymbol{\theta}_g; y_{j'}^{(i')}\right) \right\rangle \\
&= \sum_{k=1}^{K} \sum_{(j,j')} \sum_{(i,i')} \left(\alpha_{i,j} g_k\left(\boldsymbol{\theta}_g; y_j^{(i)}\right)\right) \left(\alpha_{i',j'} g_k\left(\boldsymbol{\theta}_g; y_{j'}^{(i')}\right)\right) \left\langle \nabla_{\boldsymbol{\theta}} f_k\left(\boldsymbol{\theta}_f; u^{(i)}\right) \ , \ \nabla_{\boldsymbol{\theta}} f_k\left(\boldsymbol{\theta}_f; u^{(i')}\right) \right\rangle \\
&\quad + \sum_{k=1}^{K} \sum_{(i,j),(i',j')} \left(\alpha_{i,j} f_k\left(\boldsymbol{\theta}_f; u^{(i)}\right)\right) \left(\alpha_{i',j'} f_k\left(\boldsymbol{\theta}_f; u^{(i')}\right)\right) \left\langle \nabla_{\boldsymbol{\theta}} g_k\left(\boldsymbol{\theta}_g; y_j^{(i)}\right) \ , \ \nabla_{\boldsymbol{\theta}} g_k\left(\boldsymbol{\theta}_g; y_{j'}^{(i')}\right) \right\rangle \\
&\geq \sum_{k=1}^{K} \lambda_{\min}(\mathcal{K}_{f,k} \otimes \mathbb{I}_q) \left\| \boldsymbol{\alpha} \odot g_k(\boldsymbol{\theta}_g; \boldsymbol{y}^{(\cdot)}) \right\|_2^2 + \lambda_{\min}(\mathcal{K}_{g,k}) \left\| \boldsymbol{\alpha} \odot \left(\mathbf{1}_q \otimes f_k(\boldsymbol{\theta}_f; \boldsymbol{u}^{(\cdot)})\right) \right\|_2^2 \\
&\geq \lambda_{0,f} \sum_{k=1}^{K} \left\| \boldsymbol{\alpha} \odot g_k(\boldsymbol{\theta}_g; \boldsymbol{y}^{(\cdot)}) \right\|_2^2 + \lambda_{0,g} \sum_{k=1}^{K} \left\| \boldsymbol{\alpha} \odot \left(\mathbf{1}_q \otimes f_k(\boldsymbol{\theta}_f; \boldsymbol{u}^{(\cdot)})\right) \right\|_2^2 \ ,
\end{aligned}
$$

where $\mathbb{I}_q$ denotes the $q$-dimensional identity matrix since $\alpha_{i,j} g_k\left(\boldsymbol{\theta}_g; y_j^{(i)}\right)$ varies with $j$ whereas the kernel $K_f$ does not; $\mathbf{1}_q$ is the $q$-dimensional all ones vector since for the $\alpha_{i,j} f_k(\boldsymbol{\theta}_f; u^{(i)})$ terms, for a fixed $i$, $\alpha_{i,j}$ differs with $j$ but $f_k\left(\boldsymbol{\theta}_f; u^{(i)}\right)$ stays the same; $\lambda_{\min}(\mathcal{K}_{f,k} \geq \lambda_{0,f}$; and $\lambda_{\min}(\mathcal{K}_{g,k}) \geq \lambda_{0,g}$. This completes the proof. $\qquad\square$

Note that we require $\boldsymbol{\alpha}$ is a block unit vector with $\boldsymbol{\alpha}_{i,j}$ denoting the $j$-th position in the $i$-th for convenience in dealing with the quadratic form above. Given that $T^{(3)}, T^{(4)}_{k,k'}$ terms are close to 0 in expectation, as shown in Proposition 2, we now need to show that the NTK $\mathcal{K}(\boldsymbol{\theta})$ is positive definite.

**Proposition 4.** *Given that the terms $T^{(3)}_{k,k'}$ and $T^{(4)}_{k,k'}$ can be suitably bounded close to zero by Proposition 2, we have, for any arbitrary block unit vector $\boldsymbol{\alpha}$, and some $k \in [K]$,*

$$
\left\| \boldsymbol{\alpha} \odot g_k(\boldsymbol{\theta}_g; \boldsymbol{y}^{(\cdot)}) \right\|_2^2 \geq c_1 > 0 \,, \qquad or \qquad \left\| \boldsymbol{\alpha} \odot \left(\mathbf{1}_p \otimes f_k(\boldsymbol{\theta}_f; \boldsymbol{u}^{(\cdot)})\right) \right\|_2^2 \geq c_2 > 0 \,, \qquad (44)
$$

*where $\mathbf{1}_q$ denotes the $q$-dimensional vector of all entries equal to one.*

*Proof.* Now, for the first term, for some $k \in [K]$, making use of (38), we have

$$
\alpha_{i,j} g_k(\boldsymbol{\theta}_g, \boldsymbol{y}_j^{(i)}) = \sum_{h'=1}^{m_g} w_{k,h'}^{(g)} \alpha_{i,j} \bar{g}_{h'}\left(\boldsymbol{\theta}_{\bar{g}}; y_j^{(i)}\right) = \left\langle w_{k,\cdot}^{(g)}, \alpha_{i,j} \bar{g}_\cdot\left(\boldsymbol{\theta}_{\bar{g}}; y_j^{(i)}\right) \right\rangle \,.
$$

Since the trunk net is a ReLU network, following the argument in Lemma 7.1 in (Allen-Zhu et al., 2019), with probability at least $1 - O(q e^{-\Omega(m/4L)})$, for all $(i,j)$ we have $\left\| \bar{g}_\cdot\left(\boldsymbol{\theta}_{\bar{g}}; y_j^{(i)}\right) \right\|_2 \geq 1/2$.

Recalling that $w_{k,h'}^{(g)} \sim \mathcal{N}(0, \frac{1}{K})$, taking expectation over the randomness of $w_{k,h'}^{(g)}$, we have

$$
\mathbb{E} \left\| \boldsymbol{\alpha} \odot g_k(\boldsymbol{\theta}_g; \boldsymbol{y}^{(\cdot)}) \right\|_2^2 = \sum_{(i,j)} \mathbb{E} \left\langle w_{k,\cdot}^{(g)}, \alpha_{i,j} \bar{g}_\cdot \left( \boldsymbol{\theta}_{\bar{g}}; y_j^{(i)} \right) \right\rangle^2
$$

$$
= \sum_{(i,j)} \left\langle \mathbb{E}[(w_{k,\cdot}^{(g)})^2], \alpha_{i,j}^2 \bar{g}_\cdot^2 \left( \boldsymbol{\theta}_{\bar{g}}; y_j^{(i)} \right) \right\rangle
$$

$$
= \sum_{(i,j)} \frac{\alpha_{i,j}^2}{K} \left\| \bar{g}_\cdot^2 \left( \boldsymbol{\theta}_{\bar{g}}; y_j^{(i)} \right) \right\|_2^2
$$

$$
\geq \frac{1}{2K} \|\boldsymbol{\alpha}\|_2^2 = \frac{1}{2K} \ .
$$

Hence, we have

$$
\lambda_{0,f} \mathbb{E} \sum_{k=1}^{K} \left\| \boldsymbol{\alpha} \odot g_k(\boldsymbol{\theta}_g; \boldsymbol{y}^{(\cdot)}) \right\|_2^2 \geq \frac{1}{2} \lambda_{0,f} \ ,
$$

and, with a similar argument

$$
\lambda_{0,g} \mathbb{E} \sum_{k=1}^{K} \left\| \boldsymbol{\alpha} \odot \left( \boldsymbol{1}_p \otimes f_k(\boldsymbol{\theta}_f; \boldsymbol{u}^{(\cdot)}) \right) \right\|_2^2 \geq \frac{1}{2} \lambda_{0,g} \ .
$$

As a result, we have

$$
\boldsymbol{\alpha}^T \mathbb{E} \left[ \mathcal{K}(\boldsymbol{\theta}) \right] \boldsymbol{\alpha} \geq \frac{\lambda_{0,f} + \lambda_{0,g}}{2} > 0 \ . \tag{45}
$$

The high probability version of the result can be obtained by applying Hoeffding (for cross terms) and Bernstein (for square terms) bounds on $\left\langle w_{k,\cdot}^{(g)}, \alpha_{i,j} \bar{g}_\cdot \left( \boldsymbol{\theta}_{\bar{g}}; y_j^{(i)} \right) \right\rangle^2$. That completes the analysis. $\square$

## A.5 EXPERIMENTAL DETAILS

For the optimizer we choose `Adam` (Kingma & Ba, 2014) with an adaptive learning rate schedule initialized at a learning rate $\eta_0 = 10^{-3}$. In order to generate training data for all three examples, we sample the input, denoted by $u(x)$, from a zero mean Gaussian process (GP) on a grid $\{x_l\}_{l=1}^{m} \in [0, 1]$ and generate outputs corresponding to each sampled function by solving the ODE/PDE (see (Wang et al., 2021; Lu et al., 2021) for a detailed discussion on data generation). For end-to-end training we use the deep learning framework JAX (Bradbury et al., 2018) and build our code on top of (Wang et al., 2021) for Diffusion-Reaction and Antiderivative operators and we develop our own for the Burger's equation. We now briefly outline the problems below along with the specifics of the training process for each of them.

### A.5.1 ANTIDERIVATIVE OPERATOR

The antiderivative (or simply the integral) operator corresponds to a linear operator defined explicitly by a linear ODE (initial value problem) in the unknown function $v(x) \in \mathcal{V}$, given the input $u(x) \in \mathcal{U}$, and the constant $v(0)$ for mathematical well-posedness, i.e.

$$
\frac{dv(x)}{dx} = u(x), \quad x \in [0, 1] \quad \text{s.t.} \quad v(0) = 0. \tag{46}
$$

We learn the operator mapping $u(x)$ to its corresponding integral $v(x) = G_{\boldsymbol{\theta}}(u)(x)$ for all $x \in (0, 1]$. For generating the training data, we sample the input functions from a univariate Gaussian process as outlined above and the output points randomly on the interval $[0, 1]$ and choose $n_B = 10000$ for our empirical results

### A.5.2 DIFFUSION-REACTION PDE

In this example we learn the operator mapping the input forcing function $u(x)$ to the output $v(x,t)$ for the nonlinear Diffusion-Reaction PDE given by

$$\frac{\partial v}{\partial t} = D\frac{\partial^2 v}{\partial x^2} + kv^2 + u(x), \quad (x,t) \in (0,1] \times (0,1] \quad \text{s.t.} \quad \begin{cases} v(0,x) = 0 \\ v(t,0) = 0 \\ v(t,1) = 0 \end{cases} \tag{47}$$

where $D = 0.01$ and $k = 0.01$ are constants denoting the diffusivity and reaction rate respectively. Note that in this case we are learning the operator $v(x,t) = G_{\boldsymbol{\theta}}(u)(x,t)$. For each sampled input, the PDE is solved using a backward finite-difference solver on a grid $(x,t)$ of size $(150 \times 120)$. For training, the number of input sensors is fixed at $m = 120$. The number of input samples $(n)$ is chosen to be $5000$ and $n_B = 10000$.

### A.5.3 BURGER'S EQUATION

Finally, we look at the Burger's equation benchmark similar to the one investigated in (Li et al., 2021a) with the distinction that we learn a mapping from the initial condition $v(x,0) = u(x)$ to the solution $v(x,t)$ for $(x,t) \in [0,1] \times (0,1]$

$$\frac{\partial v}{\partial t} + v\frac{\partial v}{\partial x} - \nu\frac{\partial^2 v}{\partial x^2} = 0, \quad (x,t) \in (0,1) \times (0,1]$$

$$\begin{cases} v(x,0) = u(x), & x \in (0,1) \\ v(1,t) = v(0,t) & t \in (0,1) \end{cases} \tag{48}$$

We generate the training data using a stiff PDE integrator `chebfun` (Driscoll et al., 2014) on a grid resolution of $(501, 501)$ and $p = 200$ training points sampled randomly on the solution grid.

