# OpenReview forum: "Learning Deep Operator Networks: The Benefits of Over-Parameterization"
_ICLR.cc/2023/Conference — Submitted to ICLR 2023_

### Official Review · Reviewer_4Wz6 · 2022-10-23

**Confidence:** 5
**Correctness:** 3
**Technical Novelty And Significance:** 2
**Empirical Novelty And Significance:** 2
**Recommendation:** 3

**Clarity, Quality, Novelty And Reproducibility:**

The paper is easy to read and the main results are clearly stated. However, the results are incremental and not novel enough to be published in ICLR.

**Strength And Weaknesses:**

Strengths: The analytical results on the training of operator networks seem to be new and extend the prior analysis for learning finite dimensional functions. The paper is easy to read and follow.

Weaknesses: I personally feel that this paper is merely a trivial extension of the early results of Du et al 2019 and Liu et al 2021 for finite dimensional functions to operator networks. The proof ideas and techniques based on the analysis of kernels and PL-condition are identical to those shown in the early literature. I do not see the essential difficulties in the extension except the technicalities arising from the new network architecture.


**Summary Of The Paper:**

This paper studied the convergence of gradient descent algorithm in training the DeepONet in the overparameterized regime. Specifically,  the authors proved a spectral norm bound for the Hessian of DeepONets with smooth activation functions; this bound implies convergence of GD using the RSC property. For ReLU activated DeepONets the authors showed that the corresponding neural tangent kernel at initialization is positive definite which again implies convergence of GD.

**Summary Of The Review:**

I am inclined to reject the paper due to the lack of sufficient novelty and significance.

---

> ### Author Response · Authors · 2022-11-17
> **Response**
>
> We thank the reviewer for ackowledging the analytical results. Please see below our response to the concerns:
>
>    * $\textbf{Relation\ to\ previous\ works\ for\ NNs}$: As outlined in the Introduction, there exists no optimization results on DeepONets and therefore the present proofs present a first step in understanding the optimization landscape of DeepOnets. We believe the novelty is in the model itself. Our analysis for DeepONets is simpler than that of Du et al. 2019 for neural networks.
>
>    * $\textbf{Operator\ Learning}$: The Operator Learning problem is in sharp contrast to standard supervised learning for neural networks
> and therefore we strongly believe that it warrants a separate analysis.
>
> In addition to the above, establishing the necessary conditions in Section 4 (RSC, RSS and restricted PL) is highly nontrivial for the DeepONet model.

---

### Official Review · Reviewer_BsC5 · 2022-10-24

**Confidence:** 3
**Correctness:** 3
**Technical Novelty And Significance:** 2
**Empirical Novelty And Significance:** 2
**Recommendation:** 3

**Clarity, Quality, Novelty And Reproducibility:**

Clarity
-------
As mentioned above, the paper is quite clear despite the analytic nature of the work. This makes it quite easy to read.

Quality
--------
Overall, the paper is reasonably well executed. Nothing seems to be wrong in the theoretical justification.

Novelty
---------
The paper does not feel very novel, as the techniques seem to be heavily borrowed from previous analyses and just applied out-of-the-box to DeepONets.

Reproducibility
-----------------
No code/architectural design is provided. This technically makes it hard to reproduce the experiments, but this shouldn't really matter as they are toy/sanity checking purposes only.

**Strength And Weaknesses:**

Strengths
-----------
* The paper's results are presented reasonably cleanly. This is especially nice given the heavy analytic nature of the work and greatly enhances the communication of the results.
* The paper's results all seem to be correct.
* The experimental results show that increase network width does improve the results.

Weaknesses
--------------
* The paper does not mention any of the more recent operator networks (e.g. fourier neural operators) which tend to show improved performance over DeepONets. This tends to make the analysis feel out of date.
* The overall results feel rather weak. In particular, it's not that surprising nor insightful that overparameterization would help DeepONets, especially with the wealth of literature on overparameterization.
* The proof techniques used feel like just reapplication of standard overparameterization tools. In particular, it treats the DeepONet as just a different architectural design (akin to MLPs or Resnets). However, the DeepONet primarily acts on functionals, which should necessitate something different. Is there any operator perspective to the analysis?
* The experimental results are really not necessary, don't add much, and don't really feel like a worthwhile inclusion. This is true even in comparison with general theory papers (which tend to have a weaker experimental setup), and so I would move them to the appendix and expand upon the theoretical contributions.


**Summary Of The Paper:**

This paper establishes theoretical guarantees about overparameterized DeepONets. In particular, they show that

**Summary Of The Review:**

Overall, I lean reject. In particular, my concerns are primarily that 1) DeepONets are old and better, more novel architectures have been introduced, 2) the insights given by the paper seem rather weak/intuitive, 3) the proof structure feels like a reapplication of standard techniques from the field to a different architecture.

---

> ### Author Response · Authors · 2022-11-17
> **Response**
>
> We thank the reviewer for bringing out several interesting observations. Please see our response to each of the concerns below:
>
>    * $\textbf{Relation\ to\ FNO}$: We would like to to reiterate the point presented in Introduction. “While constructing such neural operators constitutes a large body of work in itself, to the best of our knowledge, none of the preceding works have provided convergence guarantees on when and why these neural operators/operator networks and their underlying optimization algorithms converge to the global optimum"
>
> The above is true for both FNO and DeepONet which are two popular classes of Neural Operators / Operator networks that each warrant a separate analysis due to their distinct features. We do not make any claims comparing FNO with DeepONet, nor do we feel such claims to be relevant to the analysis we have presented.
>
> To summarize, in this work we have established provable guarantees for the convergence of GD for DeepONet, which is a first work of its kind. A similar analysis for FNO, while possible, is outside the scope of this work.
>
>
>    * $\textbf{Overparameterization\ helps\ DeepONets}$: We again thank the reviewer for bringing an important point to our attention. While there exists a wealth of literature on overparameterization for Neural Networks (that approximate “functions”), there exists no work that provably demonstrates the effect of overparameterization for Neural Operators / Operator Networks.
>
>
>    * $\textbf{Operator\ Perspective\ and\ Proof\ technique}$: The operator perspective is a result of the approximation guarantees of DeepONets. It is baked into the Universal Approximation Theorem for Operators in the original DeepONet paper (Lu et al., 2021). In this work we analyze the corresponding neural model for DeepONet and establish convergence guarantees for that model. The proof technique showing the RSC of the DeepONet loss and the positive-definiteness of the DeepONet-NTK are both novel and establish a first result that shows convergence of gradient descent for the corresponding operator learning problem.
>
>
>    * $\textbf{Analogy\ to\ ResNets}$: A ResNet is a fundamentally different model than DeepONet. The former approximates mapping between two finite-dimensional vector spaces. This is in sharp contrast to DeepONet which takes in as input both the input (which is a function discretized on a grid) and also the output locations (where the corresponding solution needs to be queried) and as a result learns mappings between two function spaces.
>
> This is fundamentally different from a standard ResNet. In fact, a ResNet maybe used as a sub-architecture (for the branch and/or trunk net) instead of a FNN. However, that analysis is again outside the scope of this work.
>
>    * $\textbf{Experimental Observations}$: We feel that the experiments communicate the main story fairly succinctly and complement the
> theoretical results, which, as the reviewer correctly points out, are the heart of the paper. Hence, we believe that they should be placed in the main paper to elucidate the main technical results succinctly.
>
>
>    * $\textbf{Novelty}$: We would like to further reiterate that DeepONets do not have any optimization guarantees and establishing these is the main technical result of our paper. The analysis presented in the paper builds on top of existing results for neural networks and extends these to Operator Networks.

---

### Official Review · Reviewer_gCk5 · 2022-10-25

**Confidence:** 4
**Correctness:** 4
**Technical Novelty And Significance:** 2
**Empirical Novelty And Significance:** 2
**Recommendation:** 5

**Clarity, Quality, Novelty And Reproducibility:**

This paper is a bit incremental given prior works on the NTK based convergence analysis.


**Strength And Weaknesses:**

Strength:
1. This paper proves the convergence for DeepONets with smooth activations based on the RSC condition.
2. This paper proves the convergence for DeepONets with ReLU activation based on the NTK analysis.
3. This paper conducts numerical experiments to justify the theory.

weakness:
1. The initialization of weights in Assumption 3 seems unreasonable to me. In fact, the authors consider using a small scaling for only the last layer, then the optimization on the last layer weights will dominate the training of the remaining weights. By (5), the neural network model will be very close to the linear model.
2. It is also not clear to me why the authors need to use two sections to study the convergence of DeepONets with two different types of activations. My understanding is that they can be both performed via standard NTK analysis (in the lazy training regime, there is nearly no difference between NTK and RSC based analysis).
3. In the ReLU case, it would be better to give a quantitative bound on $\lambda_{0,g}$  or $c$ in Theorem 5.1.


**Summary Of The Paper:**

This paper studies the optimization convergence guarantee of GD for training DeepONets with wide layers. Technically, the authors prove the convergence for both smooth activations and ReLU activation and show that a linear convergence rate can be achieved. The authors further present empirical results to back up the theoretical findings.

**Summary Of The Review:**

Overall, I am not satisfied with the initialization in Assumption 3, the authors may need to make the training of all layers be in the same level rather than using different initialization scaling for different layers. Besides, the contributions of this paper are not strong, especially given a large number of prior works studying the convergence of GD in the NTK regime.

---

> ### Author Response · Authors · 2022-11-17
> **Response**
>
> We thank the reviewer for raising this point and in the process helping us spot a typo in equations (4) and (5). We would like to bring clarity regarding Assumption 3:
>
>    * $\textbf{Assumption\ 3}$: Note that the scaling of the last layer is identical to the previous layers except for an additional factor of $1/K$. The $1/K$ is to take into account the $K$-predictors in the branch net in $\sum\limits_{k=1}^K f_k g_k$. Note that in the previous layers the $1/\sqrt{m}$ is explicitly multiplied with the weight matrices $W^{(l)}$, whereas for the last layer this is taken implicitly into account in the initialization variance (the $1/\sqrt{m}$ is pushed in the Gaussian to give a $1/m$ variance). We have also corrected equations (4) and (5) in light of the comment.
>
>
>
>    * $\textbf{Two\ Different\ Sections}$: As for the necessity of two different analyses we remark that both the analyses establish sufficient conditions for the convergence of GD. The positive definiteness of the NTK and the RSC of the loss are distinct and independent from each other. Since either is a sufficient condition for convergence, we believe that these warrant separate sections in the main paper.
>
>    * $\textbf{Quantitative\ Bound\ for\ }\ \lambda_{\min}$:  A bound on the minimum eigenvalues of the branch $\lambda_{0,f}$ and trunk $\lambda_{0, g}$ nets could be given in the same spirit as [Nguyen et al. 2021]. In particular,
>
>         * $\lambda_{0,f}= \min\limits_{k\in [K]} {\lambda^{(k)}_{0,f}}$
>         * $\lambda_{0,g}= \min\limits_{k\in [K]}{\{\lambda^{(k)}_{0,g}\}}$
>
> where each of the above minima are the minimum eigenvalues of the limiting tangent kernels associated with the $K$ branch and trunk net predictors respectively. These can be suitably bounded from below. While the bounds aren't sharp, these suffice for the current analysis. Since the proof follows directly from Nguyen et al. 2021, we have not reproduced it as is, and have added reference to Theorem 4.1 in Nguyen et al., 2021 in the updated draft.

---

### Official Review · Reviewer_QX4p · 2022-10-31

**Confidence:** 3
**Correctness:** 3
**Technical Novelty And Significance:** 2
**Empirical Novelty And Significance:** 2
**Recommendation:** 5

**Clarity, Quality, Novelty And Reproducibility:**

Some questions and comments about clarity:

1. Why do the learning curves (in Figure 1 and similarly 2 and 3) fluctuate more with larger width?

2. How does m affect the local convexity in the smooth activation case?

3. Any intuition why the NTK is already positive definite at initialization? Does that hold true with u being an identity function?

4. After Definition 2, L is α-RSC w.r.t. (Q, θ) not (S, θ)? And, in Theorem 4.2, the sum should be gradient of \bar(G)_t not \bar(G)_t itself?


**Strength And Weaknesses:**

Regarding the strengths, this paper
+ establishes first guarantees of GD for deep operator;
+ shows the non-trivial convergence analyses, albeit using standard analysis techniques (restricted convexity and NTK).

However, there are some drawbacks:
- The authors do not explicitly state convergence results in dedicated theorems, so it is difficulty to understand the full picture (e.g., convergence rate, role of learning rate…) In particular, while I understand how the analysis goes, I cannot see how the width (or the over-parameterization) attributes to the overall convergence.
- The paper presents two different analyses due to the smoothness without discussing limitations or why one can't apply one to the other.
- The experiments are based on Adam, an adaptive SGD, instead of GD.


**Summary Of The Paper:**

This paper provides convergence analyses of gradient descent for learning deep, over-parameterized operator networks. The authors show the guarantees two types of networks: (1) with smooth activations and (2) with ReLU activation using two different analysis techniques.


**Summary Of The Review:**

The analyses presented in this paper are non-trivial, but I think it needs some more work to be ready for acceptance (e.g., more explicit convergence results, discussions about limitations of each of the analysis techniques....)

---

> ### Author Response · Authors · 2022-11-17
> **Convergence results**
>
> We thank the reviewer for raising several important points. Our response along with corresponding updates to the draft above is detailed below.
>
>    * While the approach using NTK in Section 5 would apply to both smooth and ReLU activations, the Hessian-based approach does not directly apply to ReLU. The Hessian-based analysis, however, provides a sufficient condition for the convergence of gradient descent for the DeepONet learning problem. We have now included this as a footnote in the main paper on page 7.
>
>    * The width $m$ controls the extent of restricted strong convexity (RSC) of the loss (Theorem 4.2). We $\bf do\ not\ show\ nor\ use$ any form of strong convexity (whether local or global) for the DeepONet loss. From the outset, we consider only $\alpha$-RSC and provably demonstrate the RSC of the loss with the parameter $\alpha \equiv \alpha(m)$.
>
> To summarize, we do not guarantee local convexity for the DeepONet loss and only show α-RSC.
>
>    * Positive definiteness of the NTK at initialization:  We build the intuition in light of Assumption 3. Since the branch and trunk nets are simple feedforward networks with the parameters initialized according to Assumption 3, the NTK corresponding to each of them is positive definite. This follows directly from previous works [Du et al., 2019; Nguyen et al., 2019;]. Furthermore, since the DeepONet model is an inner product of K-predictors (the K branch and trunk net outputs) the combined NTK would "likely" be positive definite. The intuition, however, isn't sufficient and indeed we show explicitly via Theorem 5.1 (proof in in Appendix A.4) and propositions 2, 3, 4, that the resulting DeepONet NTK is positive-definite.
>
> The result holds irrespective of $u$ being an identity function or not.
>
>
>    * Indeed the loss is $\alpha$-RSC w.r.t. (Q, θ). The note above (14) is merely an elaboration of the definition of Restricted Strong Convexity (RSC) and its distinction from Strong Convexity (SC). We appreciate the reviewer for spotting out the typo in the definition of $\bar{G}_t$ and have now corrected it.

---

### Official Review · Reviewer_sgD3 · 2022-11-01

**Confidence:** 2
**Correctness:** 3
**Technical Novelty And Significance:** 3
**Empirical Novelty And Significance:** 3
**Recommendation:** 8

**Clarity, Quality, Novelty And Reproducibility:**

Clarity: Well written. Please add the Deng et al. to the related work.
Quality: Good
Novelty: There exists a previous literature on convergence analysis of DeepONet. However, the analysis technique is different.
Reproducibility: N/A

**Strength And Weaknesses:**

Strength

1. The paper is well-written and easy to read.
2. The author's convergence analysis includes both smooth activations and ReLU activations.
3. Theoretical results also match the empirical observations.

Weakness
1. Deng et al. (https://arxiv.org/abs/2102.10621) have studied the convergence analysis on DeepONets. The author should cite this work in the modified draft.

**Summary Of The Paper:**

The paper analyzes Deep operator Networks (DeepONets) regarding the optimization convergence guarantees. The author provides optimization analysis on two different networks: smooth activations and ReLU activations. Their analysis shows that the overparameterization on branch and trunk networks allows faster convergence to the global solution.

**Summary Of The Review:**

Overall, I believe it is a good paper to have the better understanding on DeepONet.

---

> ### Author Response · Authors · 2022-11-17
> **Reference added in to the Introduction**
>
> We thank the reviewer for pointing us to the reference on convergence of DeepONets for Advection-Diffusion Equation. It has now been added to the main text in the Introduction

---

### Decision · Program_Chairs · 2023-01-20

**Decision:**

Reject

**Justification For Why Not Higher Score:**

As expressed in the summary above, the paper has several positive aspects, but on the balance this meta-reviewer feels that they are outweighed by the negatives.

**Justification For Why Not Lower Score:**

Despite the fact that the paper follows a well-trodden path in analyzing over-parameterized networks, the contributions are clearly expressed and (to the best of our knowledge) new to the literature.

**Metareview: Summary, Strengths And Weaknesses:**

The paper establishes convergence guarantees on using gradient descent. for training Deep Operator Networks (DeepONets) in the over-parameterized regime. Two settings are studied: smooth activations (for which the convergence proofs are based on a restricted strong convexity/PL-type analysis), and ReLU activations (for which the proofs are based on NTK).

Opinions were split about this paper. In terms of strengths, most reviewers (including this meta-reviewer) thought that the paper was generally well-written and that the results were sound. The fact that DeepONets are becoming popular in the numerical analysis/PDE literature is also indicative of possible impact.

However, several weaknesses were also highlighted. The literature on provable training of over-parameterized networks is quite mature by now. A majority of reviewers pointed out the new proofs were more of an application of a well-worn set of analysis techniques (RSC, NTK) for a slightly more complex neural network architecture, rather than providing any special new insights. This meta-reviewer concurs with this view. Moreover, one reviewer pointed out that the effect of over-parameterization was not particularly well-illuminated. The authors' response pointed out the dependence between the RSC constant and the width, but this meta-reviewer believes that more can/should be said here about the "benefits of over-parameterization", as the paper title suggests. Finally, some concerns were raised about the assumption of non-uniform scaling of different layers at initialization, which is non-standard in practice.

Overall, this is a good paper but possibly falls short of the (very) high bar for acceptance set at this year's ICLR.

Recommendation: (weak) reject.


**Summary Of Ac-Reviewer Meeting:**

N/A